# Diurnal switches in diazotrophic lifestyle increase nitrogen contribution to cereals

Yuqian Tang [1,4], Debin Qin[1,4], Zhexian Tian [1,4], Wenxi Chen[1], Yuanxi Ma[1], Jilong Wang[1], Jianguo Yang [1], Dalai Yan[2], Ray Dixon [3] ✉ & Yi-Ping Wang [1] ✉

Uncoupling of biological nitrogen fixation from ammonia assimilation is a prerequisite step for engineering ammonia excretion and improvement of plant-associative nitrogen fixation. In this study, we have identified an amino acid substitution in glutamine synthetase, which provides temperature sensitive biosynthesis of glutamine, the intracellular metabolic signal of the nitrogen status. As a consequence, negative feedback regulation of genes and enzymes subject to nitrogen regulation, including nitrogenase is thermally controlled, enabling ammonia excretion in engineered *Escherichia coli* and the plant-associated diazotroph *Klebsiella oxytoca* at 23 °C, but not at 30 °C. We demonstrate that this temperature profile can be exploited to provide diurnal oscillation of ammonia excretion when variant bacteria are used to inoculate cereal crops. We provide evidence that diurnal temperature variation improves nitrogen donation to the plant because the inoculant bacteria have the ability to recover and proliferate at higher temperatures during the daytime.

Nitrogen is an essential element for the development and survival of living matter. Biological nitrogen fixation (BNF) carried out by bacteria and archaea plays a major role in the global nitrogen cycle, through the conversion of atmospheric nitrogen into ammonia by the enzyme nitrogenase[1]. However, chemical nitrogen fixation catalyzed by the industrial Haber-Bosch process has resulted in major anthropomorphic perturbation of the nitrogen cycle through the increased use of chemical fertilizers. Frequently, <50% of the synthetic fertilizer applied to the soil is absorbed by the crop[2] and the remainder results in major environmental issues[3,4]. On the other hand, recent estimates based upon global nitrogen budgets suggest that non-symbiotic nitrogen fixation can contribute substantially more nitrogen to cereal crops than previously envisaged[5]. This capacity has inspired scientists to enhance the development of more environmentally sustainable BNF systems as a means to replace chemical fertilizers for cereal production.

Plant-diazotroph relationships can be divided into free-living, symbiotic, and crop-associative BNF systems[6–9]. Among these, the

symbiotic nitrogen-fixation system, exemplified by the rhizobium legume symbiosis is the most efficient. In this case, bacteroids inside the plant nodule behave like a slave, whereby the flux through the ammonia assimilation pathway in the bacteroids is severely restricted to enable the release of most of the nitrogen fixed by the symbiont to the plant[10,11]. Legumes in turn, not only feed the bacteroids with photosynthesis products (malate and succinate), but also express leghemoglobin inside the nodule to protect nitrogenase from oxygen damage within the bacteroid[12]. However, the current application of plant symbiotic nitrogen fixation in agriculture is limited by the relatively narrow range of symbiotic interactions, which does not extend to economically important cereals (e.g. maize, rice, and wheat)[13]. Although, the host-range for plant-associative diazotrophic bacteria is considered to be much broader, including cereals, in some cases it remains limited by preference for specific plant genotypes[14]. In comparison with symbiotic systems, the efficiency of plant associative nitrogen-fixation is lower, due to many reasons including lack of

[1]State Key Laboratory of Protein and Plant Gene Research, School of Life Sciences & School of Advanced Agricultural Sciences, Peking University, Beijing 100871, China. [2]Department of Microbiology and Immunology, Indiana University School of Medicine, Indianapolis, IN 46202, USA. [3]Department of Molecular Microbiology, John Innes Centre, Norwich NR4 7UH, UK. [4]These authors contributed equally: Yuqian Tang, Debin Qin, Zhexian Tian. ✉e-mail: ray.dixon@jic.ac.uk; wangyp@pku.edu.cn

oxygen protection required for anaerobic/microaerobic nitrogen fixation, partnership between specific diazotrophic bacteria and plant genotypes, availability of carbon supplied by plant route exudates and lack of other nutrients in the rhizosphere[8,14]. Moreover, the flux through the ammonia assimilation pathway is not restricted in plant-associative diazotrophic bacteria. Consequently, most fixed nitrogen is used to support bacterial biomass, rather than being immediately released to the plant. Engineering diazotrophic bacteria to deliver increased levels of fixed nitrogen to crops is therefore an important goal in BNF research.

Enhancing ammonia excretion by diazotrophic bacteria requires uncoupling of nitrogen fixation from nitrogen assimilation. An important target for manipulating the flux through the ammonium assimilation pathway is glutamine synthetase (GS) encoded by the *glnA* gene, which catalyzes the formation of glutamine from ammonia and glutamate in a reaction driven by the hydrolysis of one ATP molecule per $NH_4^+$ ion assimilated[15]. Perturbing catalysis by this enzyme has dual consequences. Firstly, it provides the primary route whereby ammonia is incorporated into anabolic metabolism to synthesize amino acids and nucleic acids, and secondly, its product, glutamine, is a metabolic sensor of the nitrogen status. The latter has important consequences for nitrogen regulation since the intracellular glutamine concentration regulates the activity of the bifunctional enzyme GlnD[16], which carries out reversible post-translational modification of the PII signal transduction proteins via uridylylation (Fig. 1). This has downstream ramifications not only for transcriptional regulation of genes controlled by

the global nitrogen regulatory (Ntr) system, and for regulation of nitrogen fixation by the NifL and NifA regulatory system, but also for the activity of GS itself, which is also subject to feedback regulation by post-translational modification[17,18].

Under conditions of nitrogen sufficiency, the flux through the ammonium assimilation pathway increases the intracellular glutamine concentration to a threshold that signals nitrogen excess via GlnD, resulting in post-translational regulation of GS by the adenylyl-transferase encoded by *glnE*[19]. The resultant covalent modification of GS by AMP reduces its catalytic activity enabling a negative feedback loop on GS activity (Fig. 1). Regulation of GS activity through covalent modification provides a rapid response to changes in the nitrogen status in comparison with transcriptional regulation of *glnA* expression by the Ntr system, which operates on a slower timescale.

Disruption of GS activity to uncouple nitrogen fixation from ammonium assimilation has provided a common strategy to achieve ammonia excretion in diverse diazotrophs. Many of the initial studies targeted the activity of GS by applying positive selection for resistance to enzyme inhibitors, resulting in *glnA* mutations, for example in *Anabaena variabilis*[20,21] and *Azospirillum brasilense*[22,23]. In both cases plant growth promotion was stimulated by inoculation with the variant bacteria compared with wild-type inoculant[14,22,24,25]. A different strategy has been utilized in *Azotobacter vinelandii* to achieve controllable ammonia excretion by bringing *glnA* expression under the control of a tunable exogenous promoter, resulting in the growth promotion of both plants and microalgae[26]. More recently, synthetic biology

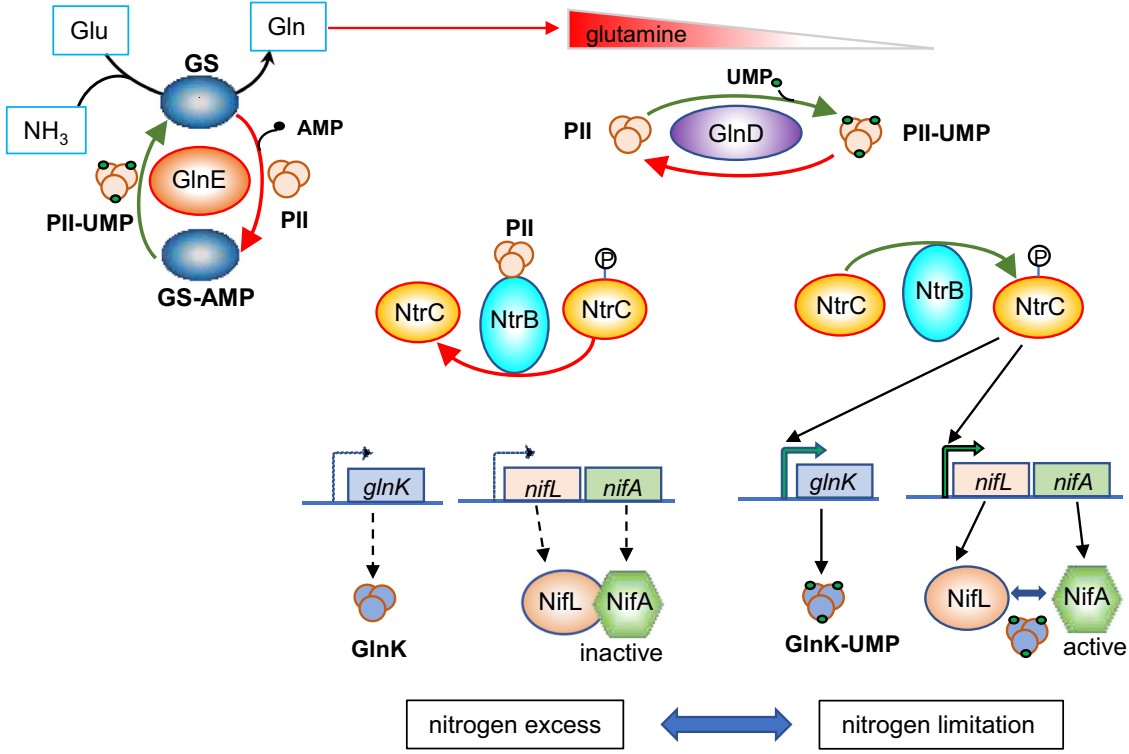

**Fig. 1 | Nitrogen regulation of nitrogen fixation in *Klebsiella oxytoca*.** The upper part of the diagram shows how the intracellular concentration of glutamine is synthesized by GS which could be modified by the bifunctional enzyme GlnE, and regulates the activity of the PII signal transduction proteins (GlnB and GlnK) as a consequence of post-translational modification by the bifunctional enzyme GlnD (UTase/UR). This has downstream consequences for transcriptional regulation of genes mediated by the nitrogen regulatory NtrBC two-component regulatory system. Under excess nitrogen conditions, when the intracellular concentration of glutamine is relatively high, the PII protein GlnB interacts with NtrB, favoring dephosphorylation of NtrC, restricting its ability to activate transcription (left side of the diagram). Under nitrogen deficient conditions, when the glutamine

concentration is low, GlnD uridylylates the PII proteins, preventing the interaction of GlnB with NtrB. As a consequence, phosphorylation of NtrC is favored, enabling activation of Ntr-regulated genes (right side of diagram). The lower part of the figure depicts the influence of nitrogen status on the regulation of nitrogen fixation by the NifL and NifA regulatory proteins. Under nitrogen-limiting conditions (right), expression of the *glnK-amtB* and *nifL-nifA* operons is activated by phosphorylated NtrC. As a consequence, GlnK is highly expressed (and also uridylylated by GlnD), preventing the formation of an inhibitory complex between NifL and NifA, thus enabling NifA to activate *nif* transcription. Under nitrogen excess conditions (left) expression of the *glnK-amtB* and *nifL-nifA* operons is limited and there is insufficient GlnK to prevent NifL from inactivating NifA.

approaches have focused on down-regulating GS through post-translation modification, using truncated versions of GlnE that constitutively adenylylate GS in *Azospirillum brasilense*[27] and *Azorhizobium caulinodans*[28]. When combined with circuits that control expression of the truncated GlnE variants this approach provides an opportunity for ammonium release in response to plant-specific signaling molecules and ultimately engineer a synthetic symbiosis. However, although such complex regulatory circuits can enable nitrogen transfer to plants, fitness defects can result in instability, requiring multicopy genetic redundancy to improve system stability[29].

Long-term stability is a common issue for diazotrophic strains in which GS activity is disrupted due to the stress imposed by nitrogen starvation and the potential for glutamine auxotrophy, combined with the energetic constraints imposed by excess nitrogenase activity that result in growth limitation. These stress conditions impose strong selection for revertants or suppressors that restore nitrogen balance and remove the growth penalty. However, previous studies with a series of *glnA* mutants of *Salmonella typhimurium* identified as being leaky glutamine auxotrophs, revealed that some of the mutant strains had similar growth rates to the wild-type when grown on ammonia as sole nitrogen source, even though the intracellular level of glutamine in these strains was significantly lower than the wild-type under these conditions[30]. This suggests that it might be possible to fine tune GS activity to achieve a sweet spot whereby the intracellular glutamine concentration signals nitrogen limitation without impeding bacterial growth, but the cell becomes blind to nitrogen availability thus preventing activation of the negative feedback loop for the Ntr system.

In this study, we have exploited one of the *S. typhimurium glnA* mutations described above[30] for the potential to engineer diazotrophs that are insensitive to excess nitrogen conditions (which we term as ammonia tolerant nitrogen fixation). Sequence analysis reveals that the mutation is located within a highly conserved region of GS in γ-proteobacteria. When this *glnA* mutant allele, encoding the GS-P95L substitution, is transferred either to *Escherichia coli* or *Klebsiella oxytoca*, decreased levels of intracellular glutamine and constitutive activation of the Ntr system are observed. Moreover, nitrogen fixation is ammonia tolerant in both *K. oxytoca* and an engineered strain of *E. coli* carrying the reconstituted *K. oxytoca nif* gene cluster. Surprisingly however, the GS-P95L substitution also confers thermo-sensitivity to GS, resulting in cold sensitive GS activity with resultant decreased rates of ammonia assimilation, enabling high levels of ammonia excretion at 23 °C, but not at 30 °C. We have exploited this thermosensitive nature of ammonia excretion by the *K. oxytoca* GS-P95L strain in relation to the seasonal temperature profile for cereal cultivation in central China. This enables temperature-dependent switching of the bacterial lifestyle during the day-night cycle and diurnal oscillation of ammonia excretion, permitting ammonia excretion at night, but enabling the inoculant to recover by resuming ammonium assimilation during the day. Our results demonstrate that this engineered *K. oxytoca* strain can deliver substantial fixed nitrogen to rice and maize when plant tests are carried out with the above diurnal temperature profiles under hydroponic conditions, with a 30–42% increase in biomass and 1.4% incorporation of $^{15}N_2$ into the total N content of maize pheophytin, in the absence of added carbon source. We demonstrate that this temperature-controlled dual lifestyle of the variant strain provides a mechanism for increased delivery of fixed nitrogen to crops, driven by environmental temperature variation.

## Results

### Characterization of a GS variant conferring altered nitrogen regulation

Guided by the phenotypes of mutations in the *glnA* gene of *S. typhimurium* (encoding GS) that exhibit growth rates similar to the wild-type strain, but have lower levels of intracellular glutamine, we selected the *glnA424* allele, originally identified in strain SK3130[30], for further investigation. Sequencing of the mutant allele revealed the amino acid substitution P95L (Fig. 2a), which is located in a highly conserved region of GS amongst γ-proteobacteria (Fig. 2b). The P95 residue is surface exposed in the structure of the *S. typhimurium* GS dodecamer[31], and is not positioned within the active site formed between adjacent subunits. However, since P95 is located in a loop in the vicinity of the active site residue D65, the leucine substitution at position 95 could potentially have an impact on catalysis (Fig. 2c).

To further investigate the properties of this mutation, we engineered the *glnA* gene in *E. coli* strain NCM3722 to encode the GS-P95L substitution (designated as strain Ec424) (see Methods). In agreement with the previous results in *S. typhimurium*, the GS-P95L substitution in strain Ec424 exhibited little impact on aerobic growth at 37 °C in the presence of excess fixed nitrogen (10 mM ammonium) (Supplementary Table 1). In contrast to the intracellular level of glutamate, which was unchanged in comparison with the wild-type strain (designated Ec), the intracellular glutamine concentration in the Ec424 strain decreased by more than twofold, which correlated with increased levels of *glnA* expression, measured with a P*glnA::lacZYA* reporter (Supplementary Table 1, aerobic conditions). This implies that the lower levels of intracellular glutamine accumulated as a result of the GS-P95L substitution, enable activation of *glnA* expression in the presence of excess ammonium, suggesting that the Ntr system in strain Ec424 is a least partially blind to the nitrogen status. To investigate if these characteristics of the GS-P95L substitution were retained under conditions appropriate to analyze the regulation of nitrogen fixation, we repeated these experiments under anaerobic conditions in L medium at 30 °C in the presence of ammonia. This analysis was carried out at the lower temperature of 30 °C since nitrogen fixation by *K. oxytoca* is temperature sensitive. Similar results were obtained, but in this case, we observed a more significant decrease in the internal glutamine level in the Ec424 strain compared with the wild-type (~4.4-fold) and a greater impact on the growth rate, which may be a consequence of the decreased glutamine levels (Supplementary Table 1, anaerobic conditions).

The Ntr system controls around 2% of genes in the *E. coli* genome[32], but these genes are differentially expressed in a sequential manner during the transition from high to low levels of glutamine, primarily dependent on the expression level and phosphorylation status of NtrC and the affinity of NtrC-P for its enhancer binding sites in the respective promoters. For example, the *glnAp2* promoter, contains a potent enhancer with two high affinity binding sites, whereas the *glnK*, *glnHp2*, and *nac* promoters have weaker enhancers[33]. Previous studies have suggested that *glnAp2*, *glnK*, *glnHp2*, *nac*, and *serA* promoters belong to different classes of NtrC-activated promoters and thus transcription from all 5 promoters was analyzed in order to assess the hierarchy of Ntr-dependent gene expression[34]. We sought to examine whether the reduction in intracellular glutamine, resulting from the GS-P95L substitution, was sufficient to promote transcriptional activation of target NtrC-dependent promoters in the presence of ammonia. qRT-PCR experiments with RNA isolated from strains grown anaerobically at 30 °C (see Supplementary Method 1), revealed that the *glnA* promoters (*glnAp1* and *glnAp2*) in the wild-type strain Ec are relatively insensitive to ammonia within the initial 1−5 mM concentration range as anticipated by the strong NtrC enhancers, and transcript levels increased in the Ec424 mutant strain as expected (Supplementary Fig. 1). We observed a ~3-fold increase of *glnA* transcript levels in Ec424 in the presence of 1 mM and 2 mM ammonium which decreased slightly to 2.5-fold when the ammonium concentration was raised to 5 mM. Similarly, activation of the *glnH* promoter in the wild-type strain was insensitive to ammonium within this concentration range, but was more strongly activated (by ~4-fold) in the Ec424 mutant at low ammonium concentrations. In contrast, the *nac* promoter was silent in the wild-type strain irrespective of the

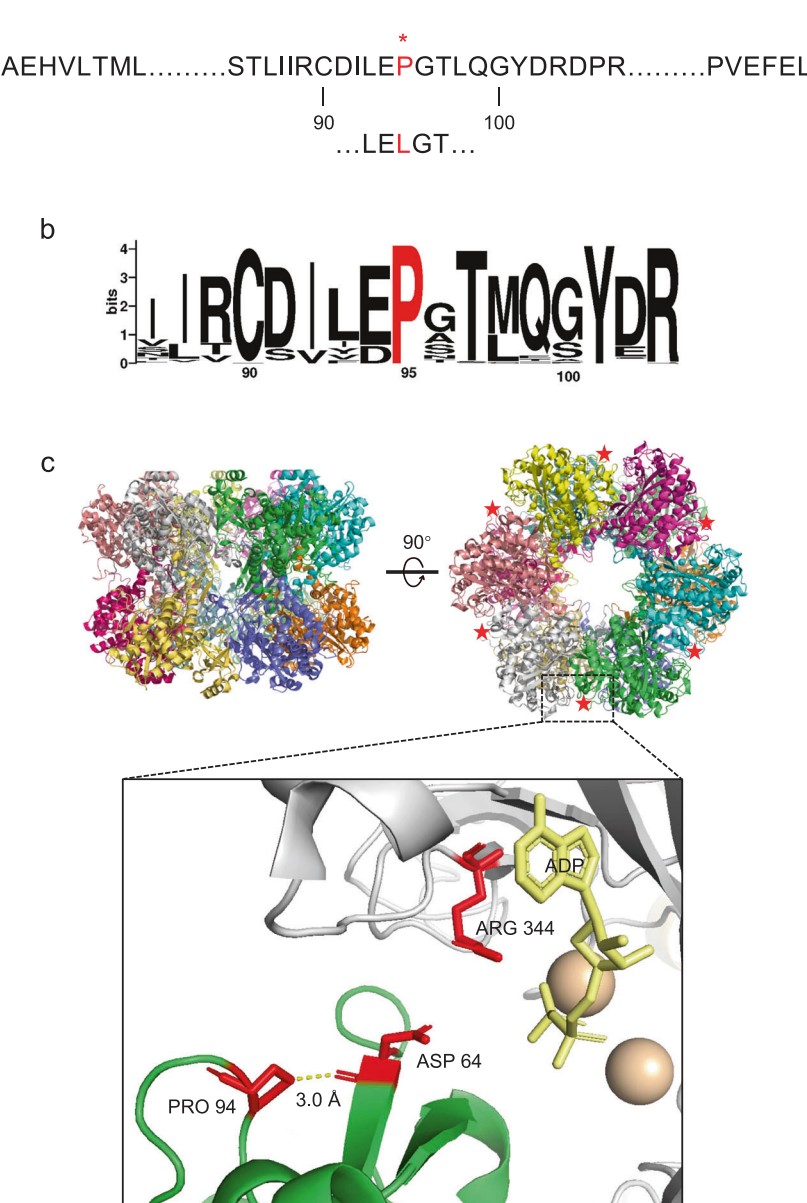

**Fig. 2 | Location of the amino acid substitution in the *S. typhimurium* variant glutamine synthetase (GS). a** Part of the amino acid sequence of GS showing the location of the P95L substitution (red lettering) encoded by the *glnA424* allele, originally identified in the *S. typhimurium* strain SK3130[30]. **b** Conserved sequence around position P95 (indicated in red) among diverse GS proteins (protein IDs provided in Supplementary Data 1). WebLogo 3 was used to draw the sequence logo. **c** The *S. typhimurium* GS dodecamer (PDB code 1F52) has 12 active sites formed between adjacent subunits within two eclipsed hexameric rings[31]. The P95 residue (red star) is surface exposed in the structure of the dodecamer and is in the vicinity (~3.0 Å) of the active site residue D65, which increases inter-subunit stability via interaction with R345. ADP in the active site is colored in yellow and the two manganese ions are shown in light brown. Note: the numbers in the expanded figure are the residues resolved in the structure, which are 1 number lower than in the protein sequence.

ammonium concentration, but large increases in *nac* transcription (~60-fold) were observed in the Ec424 mutant at 1 mM ambient ammonium concentration. This pattern was inverted for *serA* by a 7 to 50-fold decrease as expected, since the *nac* gene product, Nac, is a repressor of *serA* transcription. Similar to the *nac* promoter, activation of *glnK* transcription was inhibited by ammonium in the Ec wild-type strain, but was strongly activated by ~70-fold in the Ec424 mutant at 1 mM ambient ammonium concentration. This is important in the context of ammonia tolerant nitrogen fixation since, for example, strong activation of the *glnK* promoter by NtrC is required to express sufficient GlnK to prevent inhibition of NifA transcription by NifL in *Klebsiella oxytoca* (Fig. 1). Activation of the *K. oxytoca nifLA* promoter is also important in the context of nitrogen regulation of *nif* transcription

since this promoter contains tandem low-affinity NtrC binding sites that are only activated by high concentrations of NtrC-P[35–37]. To examine the influence of the GS-P95L substitution on *nifLA* transcription, we introduced plasmid pKU805 carrying a P*nifLA::lacZYA* fusion, into the Ec and Ec424 strains. The pattern of regulation in this case was similar to the *nac* promoter, with activation of *nifLA* transcription being strongly inhibited by ammonium in the wild-type strain, but activated in the Ec424 strain, particularly within the lower concentration range (1–2 mM ammonium by ~2.5 and ~1.3-fold respectively). Overall, these results demonstrate that the GS-P95L substitution influences the activation of NtrC-dependent promoters in the presence of ammonium to varying extents, presumably reflecting the lower glutamine level in the Ec424 strain, which in turn influences the

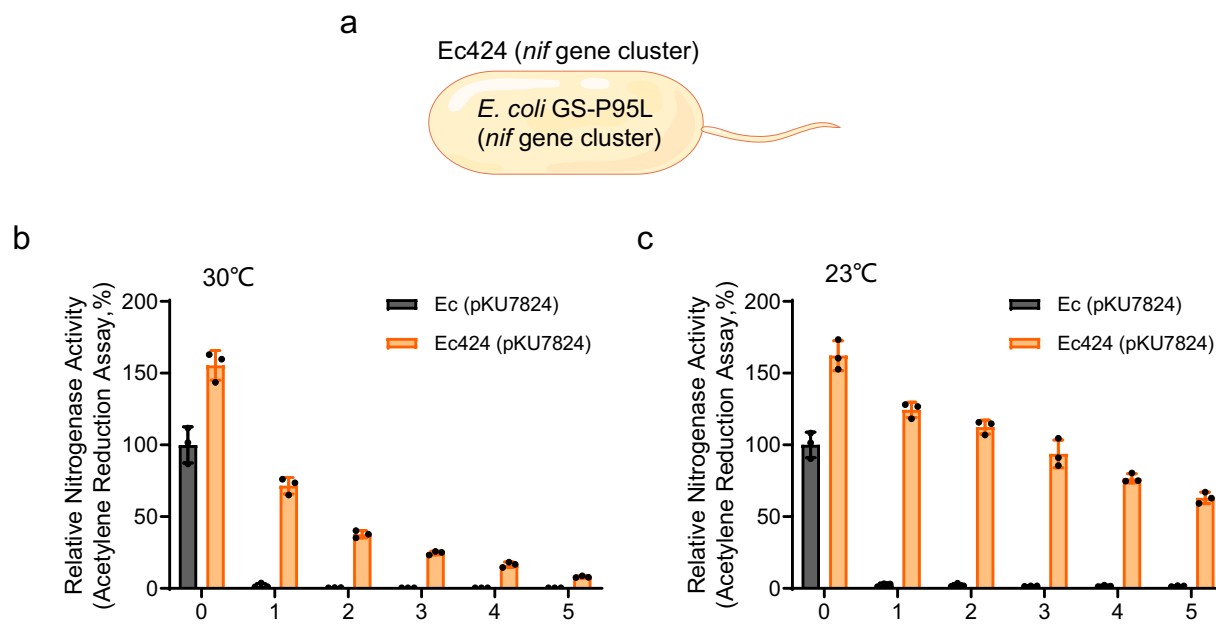

**Fig. 3 | Ammonium insensitive nitrogen fixation and ammonium excretion conferred by the GS-P95L substitution in engineered *E. coli* is responsive to temperature. a** Schematic diagram of the GS-P95L substitution in *E. coli* containing the *K. oxytoca* engineered nitrogen fixation gene cluster on plasmid pKU7824. The derivative of *E. coli* encoding the GS-P95L variant is designated as Ec424.
**b**, **c** Acetylene reduction activities determined by the *K. oxytoca nif* gene cluster on plasmid pKU7824 in either wild-type *E. coli* (Ec) (black bars) or the GS-P95L variant

of *E. coli* (Ec424) (orange bars), in response to the ammonia concentrations indicated on the *x* axes. Cultures were grown anaerobically in L medium either at 30 °C (**b**) or 23 °C (**c**) for 10 h prior to assay. Data are normalized to 100% of the nitrogenase activity of Ec (pKU7824) in the absence of ammonium (0 mM NH$_4^+$) at each temperature. Means and SDs were calculated. $n = 3$ biologically independent samples. Source data are provided as a Source Data file.

phosphorylation status of NtrC, and its ability to activate promoters dependent on enhancer binding affinities.

### The GS-P95L substitution confers ammonium tolerant nitrogen fixation and ammonia excretion in a temperature-dependent manner

Given that the P95L substitution in GS confers NtrC-dependent activation of the *nifLA* and *glnK* promoters in the presence of ammonia in *E. coli* (Supplementary Fig. 2), we sought to determine if nitrogen fixation is also ammonium tolerant in the Ec424 strain carrying the reconstituted *nif* gene cluster on plasmid pKU7824 (Fig. 3a). Nitrogenase assays (measured by acetylene reduction) were initially carried out under anaerobic conditions at 30 °C, which is conducive for nitrogen fixation, along with wild-type controls also carrying plasmid pKU7824. In contrast to the wild-type control in which nitrogenase expression and activity were prevented even at low levels of ammonia (1 mM), the Ec424 (pKU7824) strain exhibited nitrogenase activity in the presence of ammonia, although the activity declined with increasing ammonia concentrations (Fig. 3b). Interestingly, this pattern of nitrogen regulation parallels that of *nifLA* expression (Supplementary Fig. 1D), suggesting that ammonia tolerance might be limited by activation of the *nifL* promoter by NtrC-P. We also measured the external ammonia during the time course of the experiment and observed that less ammonia was consumed by the Ec424 (pKU7824) strain (Supplementary Fig. 2A). This suggests that this strain can fix nitrogen under nitrogen rich conditions, as anticipated from the acetylene reduction assays, although we cannot entirely rule out the possibility that the rate of ammonia consumption reflects differences in the growth rates of the strains. To our surprise, when these experiments were carried out at 23 °C, rather than 30 °C, we observed stronger tolerance to elevated ammonia concentrations in the Ec424 (pKU7824) strain, representing at 5 mM ammonium, about 70% of the

nitrogenase activity observed in the wild-type strain (Ec (pKU7824)) in the absence of added ammonium (Fig. 3c). This was reflected by even lower consumption of external ammonium during the time course of the experiment (Supplementary Fig. 2B).

We postulated that at the lower temperature, the GS activity of the P95L substitution could be further decreased, resulting in even further activation of the Ntr system as a consequence of lower glutamine levels signaling nitrogen deprivation. We also considered the possibility that further decreases in GS activity might uncouple ammonia assimilation from nitrogen fixation, resulting in ammonia excretion. When comparing the temperature dependency of growth, dependent upon nitrogen fixation with gaseous nitrogen as the sole nitrogen source, we observed that the Ec424 (pKU7824) strain exhibited significantly higher growth penalties compared with the wild-type strain Ec (pKU7824), as the temperature decreased (Supplementary Table 2). These temperature-dependent growth penalties were far less substantial when strains were grown aerobically in the presence of excess ammonia, suggesting that the growth defects might be associated with nitrogen fixation (Supplementary Table 3). Furthermore, at 27 °C and lower temperatures, the Ec424 (pKU7824) strain excreted ammonia. We observed a negative correlation between the amount of ammonia excreted and the growth temperature, with maximal ammonia excretion detected at 20 °C, which also correlated with the temperature-dependent decrease in growth rate (Supplementary Table 2). Thus Ec424 (pKU7824) excretes ammonia in a temperature-dependent manner.

To translate this potential ammonia delivery system from a model organism to a plant-associated diazotroph, we introduced the mutation encoding GS-P95L into the *glnA* gene of *K. oxytoca* strain M5al, thus generating a strain designated Ko424. In order to further stabilize the mutation, double nucleotide mismatches (CCA to CTG), instead of a single mismatch (CCA to CTA) were introduced at the mutation site of the *glnA* gene in *K. oxytoca* strain M5al (Fig. 4a). This strain also

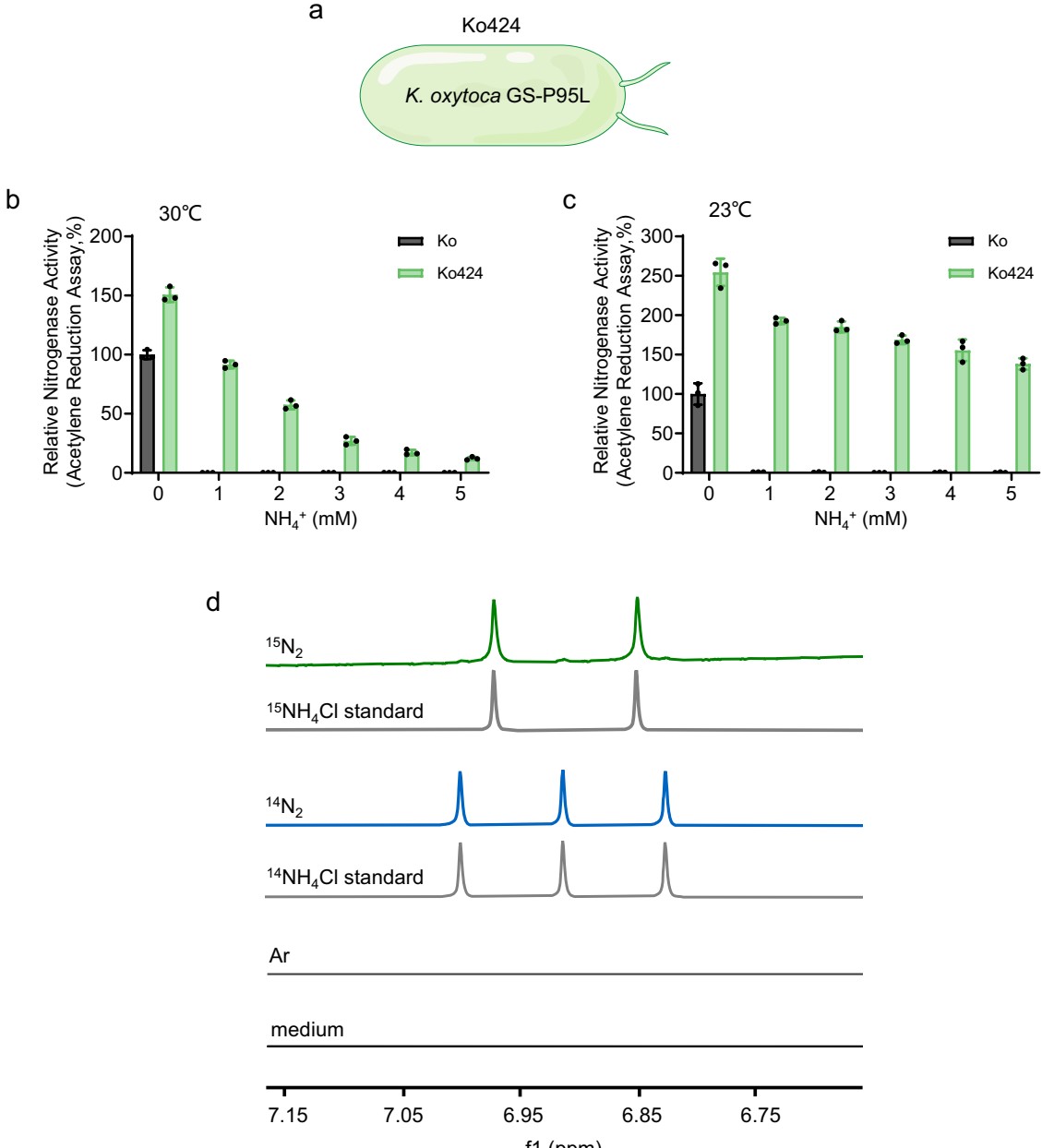

**Fig. 4 | The GS-P95L variant confers temperature-dependent nitrogen feedback regulation of nitrogen fixation and ammonium excretion in *K. oxytoca*.**
**a** Schematic diagram of the construction of the glutamine synthetase P95L substitution in *K. oxytoca*. The derivative of *K. oxytoca* encoding the GS-P95L variant is designated as Ko424. **b, c** Acetylene reduction activities of *K. oxytoca* (Ko) (black bars) or the GS-P95L variant Ko424 (green bars), in response to the ammonia concentrations indicated on the *x* axes. Cultures were grown anaerobically in L medium either at 30 °C (**b**) or 23 °C (**c**) for 10 hours prior to assay. Data are normalised to 100% of the nitrogenase activity of Ko in the absence of ammonium (0 mM $NH_4^+$) at each temperature. Means and SDs were calculated. n = 3 biologically independent samples. **d** The ammonia excreted by Ko424 is derived from nitrogen fixation. Cultures were grown in nitrogen-free medium, in which air in the anaerobic flask was evacuated with a vacuum pump and then filled with either $^{15}N_2$ or $^{14}N_2$ as the sole nitrogen source as indicated on each spectrum. A further culture was incubated with the inert gas argon (Ar) in the headspace to provide a negative control. After 72 h of anaerobic cultivation of Ko424 at 23 °C, the supernatant was collected and deuterated dimethyl sulfoxide (DMSO) (10% final concentration) and concentrated hydrochloric acid (5% final concentration) were added. To obtain the peak map of the $^1$H-NMR spectrum, samples were analyzed with a selective pulse sequence for ammonia using a high-resolution nuclear magnetic resonance spectrometer (600 MHz Bruker Avance NMR instrument)[27,38] as described in the Methods. Source data are provided as a Source Data file.

exhibited a temperature-dependent phenotype for ammonium tolerant nitrogen fixation, similar to that of the engineered *E. coli* strain Ec424 (pKU7824) (Fig. 4b, c). However, nitrogen fixation in this strain was notably more ammonium tolerant than in the *E. coli* strain at 23 °C, resulting in higher nitrogenase activities in the presence of ammonia and minimal consumption of exogenously added ammonium during the time course of the experiment. We also noted that the growth

penalty of the Ko424 strain was less severe than that of the Ec424 (pKU7824) strain at temperatures above 25 °C (compare Table 1 and Supplementary Table 2). The Ko424 mutant strain also exhibited temperature-dependent ammonia excretion, with a positive correlation between decreasing growth rate and the amount of ammonia released, resulting in maximum excretion of ~3.3 mM $NH_4^+$ at 20 °C (Table 1), compared with ~1.1 mM $NH_4^+$ in *E. coli* (Supplementary

**Table 1 | Influence of temperature on growth parameters, nitrogenase activity and ammonia excretion by Ko and Ko424**

| | Doubling time (min) | | Glutamine (mM) | | Glutamate (mM) | | Nitrogenase activity (nmol min$^{-1}$ mg$^{-1}$) | | Maximum NH$_4^+$ excretion (mM) | |
|---|---|---|---|---|---|---|---|---|---|---|
| | Ko | Ko424 | Ko | Ko424 | Ko | Ko424 | Ko | Ko424 | Ko | Ko424 |
| 33 °C | 281 ± 25 | 438 ± 19 | 7.9 ± 0.1 | 1.6 ± 0.1 | 54 ± 1 | 34 ± 1 | 10.7 ± 1.1 | 36.5 ± 1.3 | N.D. | N.D. |
| 30 °C | 255 ± 10 | 477 ± 19 | 8.8 ± 0.7 | 1.5 ± 0.2 | 58 ± 2 | 35 ± 1 | 23.3 ± 0.9 | 35.1 ± 1.4 | N.D. | 0.21 ± 0.03 |
| 27 °C | 366 ± 18 | 602 ± 24 | 6.3 ± 0.3 | 1.3 ± 0.2 | 45 ± 1 | 31 ± 1 | 19.2 ± 0.6 | 31.8 ± 1.1 | N.D. | 1.77 ± 0.05 |
| 25 °C | 487 ± 16 | 747 ± 26 | 5.2 ± 0.2 | 1.0 ± 0.1 | 38 ± 1 | 28 ± 1 | 14.6 ± 0.7 | 28.4 ± 0.9 | N.D. | 2.32 ± 0.11 |
| 23 °C | 614 ± 27 | 908 ± 29 | 4.4 ± 0.2 | 0.7 ± 0.2 | 33 ± 1 | 26 ± 1 | 9.7 ± 1.3 | 24.8 ± 1.7 | N.D. | 2.98 ± 0.08 |
| 20 °C | 783 ± 35 | 1375 ± 47 | 3.6 ± 0.1 | 0.5 ± 0.1 | 27 ± 1 | 23 ± 1 | 4.4 ± 0.6 | 20.9 ± 1.2 | N.D. | 3.32 ± 0.15 |

Note that the ammonia excretion values do not correspond to the final ammonium concentrations in Supplementary Fig. 2C and D, since maximum ammonium excretion is observed after a much longer incubation period. Ko, inoculation with the wild-type strain of *K. oxytoca*; Ko424, inoculation with the GS-P95L mutant strain of *K. oxytoca*. The means and SDs were calculated based on at least three biological replicates. N.D. indicates not detectable. Source data are provided as a Source Data file.

Table 2). Time courses of ammonia excretion in both *E. coli* and the Ko424 strain cultured at 23 °C revealed that peak rates of ammonium accumulation occurred after maximum nitrogenase activities had been reached (Supplementary Fig. 3). The level of ammonia excretion also correlated with temperature-dependent decreases in the intracellular glutamine concentration in the Ko424 strain (from ~ 1.6 mM at 33 °C to ~0.5 mM at 20 °C) (Table 1 and Supplementary Fig. 4). This is consistent with increased ammonia tolerant nitrogen fixation at the lower temperatures resulting from signals of nitrogen deficiency, as a consequence of low glutamine levels. These results also imply that the catalytic activity of GS-P95L decreases significantly at the lower temperatures, thus decreasing the rate of ammonia assimilation by GS, resulting in ammonia excretion. In contrast to the differences in glutamine levels between the wild-type and mutant strains (~5.9-fold at 30 °C and ~6.3 fold 23 °C), glutamate concentrations were less than twofold lower in strain Ko424 compared with the wild-type strain Ko, consistent with maintenance of the glutamate pool irrespective of nitrogen deprivation[30]. Overall, these results demonstrate that critical uncoupling between nitrogen fixation and nitrogen assimilation occurs in the Ko424 strain, resulting in ammonia excretion at temperatures lower than 30 °C, most likely as a consequence of decreased activity of the GS-P95L variant at the lower temperatures.

To demonstrate whether the ammonia detected in the culture supernatants originates from nitrogen fixation during anaerobic growth or for example, from turnover of other intracellular nitrogen sources, we incubated Ko424 in N-free L medium under an atmosphere of $^{15}$N$_2$ gas and demonstrated with $^1$H-NMR[38] that the supernatant exhibited the same predominant doublet as the $^{15}$NH$_4$Cl standard (Fig. 4e). Thus, we conclude that the ammonia in the medium is derived de novo from nitrogen fixation by Ko424.

As both the expression and activity of GS are regulated in response to the nitrogen status, we considered the possibility that feedback regulation of the activity of the P95L variant via post-translational modification by the adenylyltransferase GlnE[17,18], might be responsible for the ammonia excretion phenotype. In order to investigate this, we constructed a derivative of strain Ko424 in which the *glnE* gene was deleted (designated Ko424Δ*glnE*). However, no detectable changes in ammonia excretion were observed in the Ko424Δ*glnE* strain when compared with the parental strain Ko424 at 23 °C (Supplementary Fig. 5). Thus, either the P95L variant of GS is not adenylylated by GlnE, or the intracellular glutamine concentration in this variant is not sufficient to activate the feedback mechanism, which seems highly likely given the low internal glutamine levels in Ko424.

### Biochemical properties of the P95L variant of glutamine synthetase

To characterize the *K. oxytoca* GS-P95L enzyme in vitro, we expressed N-terminal his tagged derivatives of the wild-type and variant proteins in a *glnA glnE* double mutant of *E. coli* BL21 and purified both proteins

by nickel affinity chromatography (see Supplementary Method 2). Measurements of GS biosynthetic activity using a phosphate release assay (see Supplementary Method 3) revealed that the $K_m$ for glutamate for the wild-type GS enzyme varied with temperature (between ~11 mM at 20 °C to ~3.5 mM at 37 °C) in agreement with the published data at 25 °C for the *E. coli* GS enzyme (~4 mM[17]) (Supplementary Fig. 6A and Supplementary Table 4). (We note that the amino acid sequences of GS from both *K. oxytoca* and *E. coli* are very similar to each other, for details see Supplementary Fig. 7). The kinetic properties of the GS-P95L enzyme with respect to glutamate are clearly different, although it was not possible to accurately determine $K_m$ values at the different temperatures, possibly because the enzyme activities were low (Supplementary Fig. 6B). At saturating glutamate concentrations (250 mM L-glutamate for details, see Methods), the apparent $K_m$ for NH$_4^+$ of the wild-type enzyme did not appear to differ substantially from 20 °C to 37 °C, but differential decreases in enzyme activity were observed between the wild-type and variant enzymes at 30 °C compared with 23 °C (Supplementary Fig. 6C and D). Overall, these data suggest that the GS-P95L enzyme is more temperature sensitive than wild-type GS in vitro, although the mechanistic basis for temperature sensitivity is not resolved.

### Influence of temperature on ammonium donation by Ko424 to the eukaryotic alga *Chlorella sorokiniana*

We anticipated that ammonia excretion by Ko424 might support the growth of recipient organisms under nitrogen-limiting conditions in a temperature-sensitive manner. Ammonia excretion by Ko424 could be visualized by anaerobic co-culture with the photosynthetic eukaryotic alga *C. sorokiniana* on nitrogen-free solid L media, when grown in an illuminated cabinet with glucose as carbon source. Only trace growth of *Chlorella* adjacent to the Ko424 strain of *K. oxytoca* was observed at a constant temperature of 30 °C (Supplementary Fig. 8A) in contrast to incubation at 23 °C, where *Chlorella* adjacent to Ko424 turned green, as an indication of obvious growth. However, wild-type *K. oxytoca* enabled only trace growth of *Chlorella* at 23 °C, with no evidence of photosynthesis (Supplementary Fig. 8B). In agreement with the plate phenotype, the total amount of chlorophyll *a* synthesized by *Chlorella* adjacent to Ko424 was significantly higher at 23 °C than at 30 °C (Supplementary Fig. 8C). This implies that Ko424 expressing the P95L substitution in GS can support the growth of a eukaryotic alga in a temperature-dependent manner, presumably by providing fixed nitrogen at 23 °C.

We considered the possibility that oscillating temperature shifts might enable more nitrogen donation by the bacteria to *Chlorella* if the donor is given time to recover and proliferate at 30 °C, rather than being incubated continuously at 23 °C, since constitutive ammonia excretion at this temperature is likely to lead to severe nitrogen starvation in the bacteria. When we subjected the co-culture to 12 hr oscillating temperature shifts between 30 °C and 23 °C, the growth of

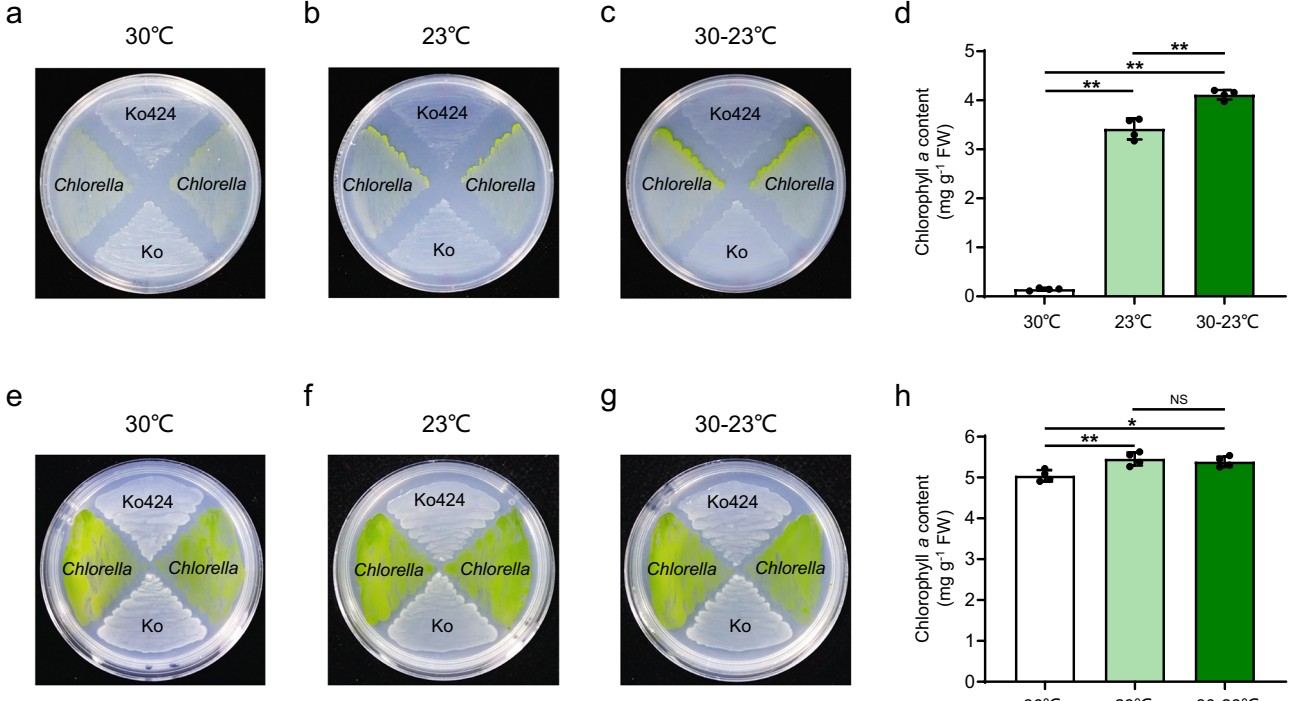

**Fig. 5 | Analysis of Ko424 as ammonia donor for growth of *Chlorella sorokiniana*. a–c** Growth promotion of the eukaryotic algae *Chlorella sorokiniana* by Ko424 and Ko in N-free L agar at constant 30 °C, constant 23 °C or with 12 h temperature shifts between 30 °C and 23 °C (30 °C–23 °C), respectively. Strains were streaked in a fan-like arrangement on the plates with *C. sorokiniana* alternating between Ko424 and Ko as indicated. Plates were incubated in transparent anaerobic tanks and grown under light illumination for 7 days. **d** Chlorophyll *a* content of *C. sorokiniana* adjacent to Ko424 at constant 30 °C, constant 23 °C or with 12 hr temperature shifts between 30 °C and 23 °C (30 °C–23 °C), respectively. The means and SDs were calculated. *n* = 4 biologically independent samples. Statistical significance is indicated as **$P \le 0.01$ analyzed using one-way ANOVA with Student's *t*-test. *P* values from left to right: 0.000000094 (30 °C vs 23 °C), 0.00000000029 (30 °C vs 30–23 °C), and 0.00107 (23 °C vs 30–23 °C). **e–g** Growth of the eukaryotic algae *Chlorella sorokiniana*, Ko424, and Ko in L agar supplemented with 3 mM ammonium at constant 30 °C, constant 23 °C or with 12 hr temperature shifts between 30 °C and 23 °C (30 °C–23 °C), respectively. Plates were incubated in transparent anaerobic tanks and grown under light illumination for 7 days. **h** Chlorophyll *a* content of *C. sorokiniana* adjacent to Ko424 at constant 30 °C, constant 23 °C or with 12 hr temperature shifts between 30 °C and 23 °C (30 °C –23 °C), respectively. The means and SDs were calculated. *n* = 4 biologically independent samples. Statistical significance is indicated as **$P \le 0.01$ analyzed using one-way ANOVA with Student's *t*-test. NS, non-significant. *P* values from left to right 0.0087 (30 °C vs 23 °C) and 0.011 (30 °C vs 30–23 °C). Source data are provided as a Source Data file.

*Chlorella* was apparently enhanced, resulting in an increase in the chlorophyll *a* content of *Chlorella* adjacent to Ko424 compared with incubation at constant 23 °C (Fig. 5a–d). In contrast *Chlorella* grew normally in the presence of ammonia with high chlorophyll levels, irrespective of the temperature (Fig. 5e–h). The importance of fluctuating temperature shifts in supporting nitrogen donation by Ko424 is further explored below.

**Fixed nitrogen delivery to plants by *K. oxytoca* Ko424**
When considering the potential benefit of strain Ko424 in agriculture as a plant associative diazotroph that exhibits thermosensitive ammonia excretion, we examined the detailed seasonal temperature profile for cereal plantations in central China. Interestingly, the average daily temperature shift between day and night is from 30 °C to 23 °C during the summer months (Supplementary Table 5), which fits well with the temperature profile of ammonia excretion by Ko424. As a prelude to performing plant experiments, we first examined the stability of the GS-P95L phenotype in both the Ec424 (pKU7824) and Ko424 strains after a more prolonged period of incubation at 23 °C. (Nucleotide substitutions encoding this GS variant in the different strains is shown in Supplementary Fig 9A). We observed in both cases that after reaching stationary phase, the ammonia excreted into the culture medium remained constant for up to 144 hr (Supplementary Fig. 9B). This implies that although the bacteria are subjected to severe nitrogen stress under these conditions, escape mutants that acquire

the ability to reassimilate the excreted ammonia do not arise frequently in the population. In order to further consider the potential use of the bacteria as inoculants in the environment, we carried out stability studies of Ko424 after growth for up to 40 generations at either 23 °C or 30 °C (see Supplementary Method 4). In each case we screened 400 individual colonies of the mutant strain and observed that all of these remained competent to support the growth of *C. sorokiniana* (Supplementary Fig. 9C). Thus, reversion of the mutant phenotype was not detectable for up to 40 generations of bacterial cell proliferation.

To investigate the potential of Ko424 for plant growth promotion under conditions that simulate the summer temperature profile in central China, we carried out coculture experiments with *japonica* rice grown under hydroponic conditions with 30 °C day and 23 °C night temperature shifts (see Supplementary Method 5 and 6), representing the temperature profile in the air as well as in the hydroponic system (Supplementary Fig. 10). After an initial growth period of 4 days, the root system of the rice was incubated for 1 hour with a suspension of inoculant bacteria at a bacterial cell density of $10^9$ cells per mL. Plants were then transferred back to the hydroponic system and grown for a further 12 days prior to harvesting. In the absence of any added carbon or nitrogen, Ko424 increased the apparent plant biomass by 12% when compared with wild-type *K. oxytoca* and also significant increases in comparison with the uninoculated control (CK), or a derivative of the wild-type strain carrying a complete deletion of the *nif* gene cluster

(KoΔ*nif*) (Supplementary Fig. 11A and B) This plant growth promotion by Ko424 was also reflected in an increase in the total nitrogen content (18%) of rice plants in comparison with the wild-type strain (Supplementary Fig. 11C).

Similar plant growth experiments as described above were carried out with maize, but in this case as more maize cultivars were available to us, we performed preliminary screening with eight different maize inbred lines, in order to determine if Ko424 has a preference for specific plant genotypes. Maize plants were grown under hydroponic conditions with the same day-night temperature shifts used for the rice experiments, with no added carbon source (Supplementary Fig. 12). In general, wild-type *K. oxytoca* or the KoΔ*nif* mutant provided little impact on maize growth compared with the uninoculated control, but three maize inbred lines responded positively to Ko424, exhibiting increases in dry weight of 14% (with B73), 20% (with Fu8701) and 29% (with 93-63) in comparison with the wild-type strain (Supplementary Fig. 12).

Subsequent experiments focusing on the maize 93-63 line, suggested that all the *K. oxytoca* strains reduced the oxygen concentration in the hydroponic system to microaerobic levels (-1–2%), which are appropriate for nitrogen fixation by this diazotroph. Moreover, a GFP-expressing version of wild-type *K. oxytoca* colonized the plant root surface, including the lateral roots and root hairs, suggestive of an efficient plant-microbe interaction (Supplementary Fig. 13). Guided by these findings, detailed experiments were carried out with the maize 93-63 inbred line in the absence of added nitrogen source, again applying temperature shifts between day (30 °C) and night (23 °C). Examples of the plant growth experiments are shown in Fig. 6. An independent iteration of these experiments was also carried out as a measure of reproducibility (Supplementary Fig. 14). The values reported below represent the range between the two independent iterations. After 9 days of bacterial inoculation, a small but statistically significant biomass increase was observed with *K. oxytoca* wild type (Ko), when compared with non-inoculated controls. However, as a similar increase was observed with the KoΔ*nif* mutant control, this cannot be a consequence of BNF. A more significant increase in dry weight was observed with Ko424, which increased plant biomass by -30–42% when compared with Ko (Fig. 6a, b and Supplementary Fig. 14A). When the nitrogen content of the plants was compared under the same conditions, only the Ko424 strain resulted in a significant increase in N content (Fig. 6c and Supplementary Fig. 14B). Compared with wild-type Ko, inoculation with Ko424 increased the nitrogen content by -34–38%. When similar experiments were carried out at a constant temperature of 23 °C, smaller increases in dry weight and nitrogen content were observed with Ko424, when compared with every other control (Fig. 6d, e, Supplementary Fig. 14C and D). Notably, the N content of the maize line decreased to -22–23% when co-cultured with Ko424 at 23 °C (11–16% less than observed with the diurnal 30 °C –23 °C temperature shift). Moreover, no significant increases in dry weight or nitrogen content were observed in comparison with the other controls when Ko424 was inoculated on maize grown at a constant 30 °C (Fig. 6f, g, Supplementary Fig. 14E and F). Overall, these results indicate that Ko424 can promote better growth and higher N content of maize line 93-63 when plants are provided with a diurnal 30 °C–23 °C temperature shift, compared with plants grown constantly at either 23 °C or 30 °C.

To estimate the contribution of nitrogen fixation to plant growth we carried out [15]N dilution experiments with the maize 93-63 inbred line grown in the presence of 0.5 mM [15]N labeled nitrate, again applying temperature shifts between day (30 °C) and night (23 °C). An increase in dry weight was observed in the presence of Ko424 under these conditions, when compared with the other controls (Fig. 7a). Data from [15]N isotope dilution experiments, calculated as the percentage of N derived from the atmosphere (%NF), showed that after 12 days of bacterial inoculation, Ko424 provided -14% of plant N through BNF in comparison with reference plants inoculated with the non-nitrogen fixing strain KoD*nif* (Table 2). When the [14]N in the seed is considered and subtracted, Ko424 provided ~26% of plant N through BNF in comparison with reference plants inoculated with the non-nitrogen fixing strain KoD*nif* (Table 2). Notably, although the wild-type Ko strain is a diazotroph, it did not result in significant [15]N dilution, confirming that this strain does not donate nitrogen to maize via BNF.

To demonstrate that nitrogen incorporated into plant biomass is directly derived from nitrogen fixation by Ko424, we incubated either maize or rice plants in coculture with the bacteria in gas-tight bags, in which 50% of the atmosphere was displaced with pure [15]$N_2$, supplemented with 1% $CO_2$. After 8 days incubation, we determined the [15]N content of the plant-specific chlorophyll derivative pheophytin and observed 1.4% incorporation of [15]N atoms in the total N of pheophytin for Ko424-inoculated maize (Fig. 7b, c), and 0.4% incorporation in the total N of pheophytin for Ko424-inoculated rice (Supplementary Fig. 15), both of which are statistically significant compared with the KoD*nif* inoculant and the Ko-inoculant. Although the level of incorporation of [15]N into plant pheophytin was an order of magnitude lower than that demonstrated in the [15]N dilution experiments, we postulate that this is a consequence of the artificial physiological environment imposed by incubating plants in a sealed environment[39] in the absence of added carbon source, where in our conditions plant biomass, chlorophyll content, nitrate assimilation and the ability of plant exudates to support the growth of *K. oxytoca* are significantly compromised (Supplementary Fig. 16). These results therefore confirm that fixed nitrogen derived from dinitrogen gas is directly transferred to plants cultivated in the presence of Ko424. However, given the limitations of both the [15]N dilution and the [15]N incorporation techniques, it is difficult to precisely quantify the amount of nitrogen directly transferred to the plants via BNF.

## Discussion

Amongst many options for improving associations between diazotrophic soil bacteria and crops, engineering ammonia excretion has been given the most prominence due to its potential for increased nitrogen delivery. Although this provides a basic mimic of the legume-rhizobium symbiosis, where the flux through the ammonia assimilation pathway is more restricted to enable release of most of the nitrogen fixed by the symbiont[10,11] the physiological impacts of ammonia excretion by associative diazotrophs in the root microbiome are completely different to those in the legume bacteroid, where optimal conditions for nitrogen fixation and ammonium delivery are provided by the nodule environment. The GS variant we have characterised here, which conditionally enables both ammonia insensitive nitrogen fixation and ammonia release as a consequence of thermo-sensitive feedback regulation, provides a strategy for temporal regulation of ammonia excretion in response to average day and night temperature fluctuations in agricultural environments. At day-time temperatures of 30 °C or above, GS activity is sufficient to enable coupling of nitrogen fixation to ammonia assimilation, thus restricting excretion of nitrogen and enabling the bacteria to proliferate. Conversely at night-time temperatures, the decrease in GS activity disfavors ammonium assimilation, enabling ammonia excretion and the lower levels of intracellular glutamine support ammonia tolerant nitrogen fixation. Our studies reveal that thermo-mediated switching of ammonia excretion on and off during the day-night cycle affords considerable benefit to the bacteria, including escape from long-term nitrogen depletion and the ability to increase biomass. Diurnal-mediated switching of GS activity provides a paradigm for an externally driven natural oscillator, which in response to ambient temperature generates an oscillatory flux of glutamine through a single amino acid change in GS that is genetically stable and influences the regulation of an array of genes. The simplicity of this environmentally responsive metabolic oscillator contrasts strongly with strategies

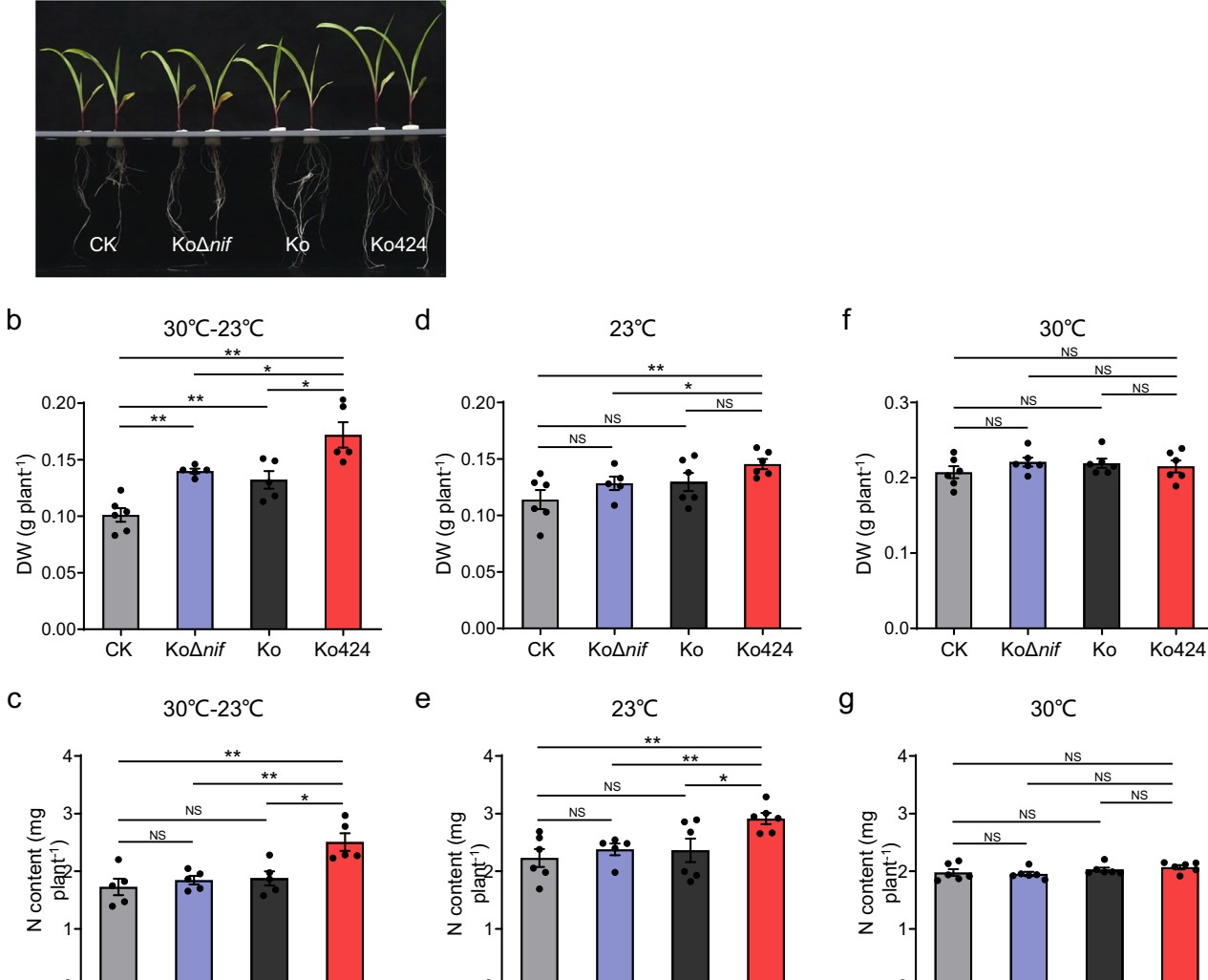

**Fig. 6 | Growth promotion by Ko424 inoculated on maize plants. a** Influence of inoculation with wild-type or mutant strains of *K. oxytoca* on the growth phenotype of the maize inbred line 93-63 after 9 days in the absence of added nitrogen source. CK indicates no bacteria were added; KoΔ*nif*, inoculation with the *nif* gene cluster deletion mutant of *K. oxytoca*; Ko, inoculation with the wild-type strain of *K. oxytoca*; Ko424, inoculation with the GS-P95L variant strain of *K. oxytoca*. **b, c** Dry weight and nitrogen content, respectively, of maize after 9 days inoculation with or without the wild-type and mutant strains of *K. oxytoca* with 12 h temperature shifts between day (30 °C) and night (23 °C) (30 °C−23 °C). DW, dry weight. N content, nitrogen content. **d, e** Dry weight and nitrogen content, respectively, of maize after 9 days inoculation with or without the wild type and mutant strains of *K. oxytoca* at constant 23 °C. **f, g** Dry weight and nitrogen content, respectively, of maize after 9 days inoculation with or without the wild type and mutant strains of *K. oxytoca* at constant 30 °C. The means and SEMs were calculated. $n = 5–6$ biologically independent samples. Statistical significance is indicated as *$P \leq 0.05$, **$P \leq 0.01$ analyzed using one-way ANOVA with Student's *t*-test. NS, non-significant. *P* values from left to rignt in **b**: 0.00031 (CK vs KoΔ*nif*), 0.0099 (CK vs Ko), 0.00022 (CK vs Ko424), 0.021 (KoΔ*nif* vs Ko424) and 0.018 (Ko vs Ko424). *P* values from left to rignt in **c**: 0.0061 (CK vs Ko424), 0.0047 (KoΔ*nif* vs Ko424) and 0.012 (Ko vs Ko424). *P* values from left to right in **d**: 0.008 (CK vs Ko424) and 0.044 (KoΔ*nif* vs Ko424). *P* values from left to right in **e**: 0.0041 (CK vs Ko424), 0.0043 (KoΔ*nif* vs Ko424) and 0.034 (Ko vs Ko424). Source data are provided as a Source Data file.

required to design and build de novo synthetic oscillators, which involve the assembly and fine tuning of complex genetic circuits, often with coupled positive- and negative-feedback loops[40–42]. When considering the stringent regulations associated with the release of genetically modified microorganisms in agriculture, single amino acid substitutions conferring diurnal changes in lifestyle, provide significant advantages over more elaborate synthetic engineering.

Diurnal modulation of ammonia excretion in response to light-dark temperature fluctuations, has some features in common with unicellular cyanobacteria whereby nitrogen fixation in the dark period is temporally separated from oxygenic photosynthesis in the light[43]. Many soil diazotrophs can only fix nitrogen under microaerophilic

conditions that are compatible with the oxygen sensitivity of nitrogenase. As strong diurnal changes in oxygen concentration can occur in the vicinity of aquatic plant roots in response to the photoperiod[44–46], confining high nitrogenase activity to the night period, when temperatures are relatively low, confers the potential to provide appropriate oxygen tensions to support nitrogen fixation. Although we did not observe circadian-related oxygen concentration fluctuations in our hydroponic experiments (Supplementary Fig. 13B), this may relate to differences between maize and aquatic plants, or other limitations in our model system.

The efficiency of plant-associative nitrogen fixation is often plant variety dependent and, in this study, amongst 8 different maize inbred

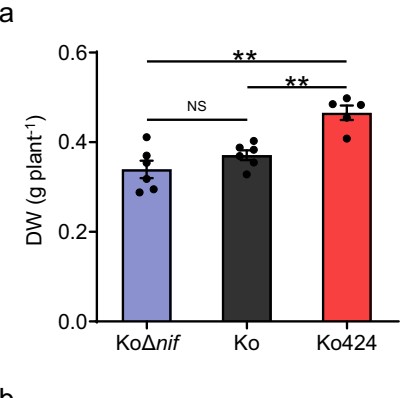

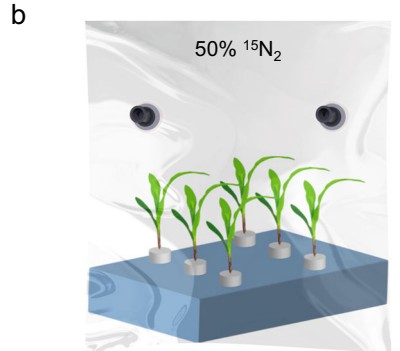

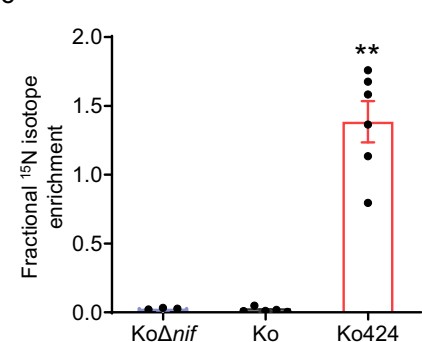

**Fig. 7 | $^{15}$N isotope labeling experiment with maize inbred line 93-63 and different bacterial inoculants. a** $^{15}$N-dilution experiments with maize inbred line 93-63 and different bacterial inoculants. 0.5 mM $^{15}$NO$_3^-$ was added at the time of inoculation. Dry weight (DW) of maize after 12 days inoculation of maize with either wild type or mutant strains of *K. oxytoca*. The means and SEMs were calculated. *n* = 5–6 biologically independent samples. Statistical significance is indicated as **$P \leq 0.01$ analyzed using one-way ANOVA with Student's *t*-test. NS, non-significant. *P* values from left to right: 0.00088 (KoΔ*nif* vs Ko424) and 0.00071 (Ko vs Ko424). **b**, **c** Direct $^{15}$N incorporation into maize from $^{15}$N$_2$. **b** Schematic diagram of the maize hydroponics system for $^{15}$N isotopic analysis. **c** Determination of fractional enrichment of the $^{15}$N isotope into pheophytin from maize inoculated with KoΔ*nif*, Ko and Ko424. 50% of the co-culture gas was displaced with $^{15}$N$_2$ gas and the final oxygen concentration in the gas mixture was 20%. 1% CO$_2$ was added daily. The results were subtracted from the fractional $^{15}$N isotope enrichment of the uninoculated control. Asterisks indicate a statistically significant difference relative to Ko. The means and SEMs were calculated. *n* = 6 biologically independent samples. Statistical significance is indicated as **$P \leq 0.01$ analyzed using one-way ANOVA with Student's *t*-test. NS, non-significant. Two-sided *P* value: 0.0000036 (Ko vs Ko424). Source data are provided as a Source Data file.

lines screened for their response to inoculation with Ko424, one line (93-63) responded most significantly. In this case, Ko424 increased the nitrogen content of the plant by 34% as a consequence of inoculation, again emphasizing the importance of the plant genotype in plant-associative BNF systems. In contrast, the increased nitrogen content of rice plants provided by Ko424 was only 18%. However, this could be due to the lack in screening different rice varieties, as *Japonica* may not be the preferred genotype for Ko424. Also, in contrast to maize, which is a C4 plant, carbon availability in root exudates required for fueling the energy-intensive process of BNF may differ between rice (a C3 plant) and maize. Indeed, when plant root exudates were used as the sole carbon source for bacterial cultivation, Ko424 cell proliferation was lower in exudates from *Japonica* rice roots, when compared with maize root exudates (see Supplementary Method 7) (Supplementary Fig. 17). These differences emphasize the many parameters that have to be considered in order to achieve more efficient plant associative nitrogen fixation. Nevertheless, our studies reveal the potential for exploiting engineered diazotrophic bacteria that have the capacity to switch between ammonia excretion and ammonium assimilation in response to environmental cues. We demonstrate that a thermally driven switch, operating within temperature profiles in natural environments, enables the diazotroph to act as a bacteriod-like ammonia donor during the cooler night period with a substantial growth sacrifice, but as the temperature rises during the day, allows free-living diazotrophic behavior that favors cell recovery and proliferation (Fig. 8). This dual lifestyle provides more efficiency than constant ammonia excretion (when the bacteria are incubated continuously at 23 °C), exemplified by an increase of the nitrogen content of maize by ~11–15%. We acknowledge that our experiments so far have been performed using cocultures under artificial conditions in the laboratory or greenhouse and may not be representative of real-world conditions, where the bacteria may experience competition with other organisms in the rhizosphere and thermal profiles will vary, with relatively slow temperature changes occurring in the natural environment compared with the abrupt temperature shifts imposed in our experiments. Nevertheless, since reversion of the ammonia excretion phenotype was not detectable during 40 cycles of bacterial cell proliferation (Supplementary Fig. 9) this phenotype is likely to be stable in the wild.

Although it may be feasible to fine tune the thermal switch to operate within the diverse temperature profiles observed in agricultural environments, we recognize that temperature is only one of many signals that could be utilized to drive the dual lifestyle switch. For example, optogenetic switches[47] or bacterial circadian oscillators[48] could be employed to mediate diel control of nitrogen donation by plant-associated diazotrophs. Alternatively, if a strategy involving plant control of associative nitrogen fixation is envisaged[49], the crop-microbe signaling circuit could potentially be linked to the plant circadian clock, thus enabling 24 h temporal control of ammonia donation. Overall, the advantages of the dual diazotrophic lifestyle elaborated in this study provide a blueprint for further improvement of plant-associative nitrogen fixation.

## Methods

### Bacterial strains and culture conditions

The bacterial strains used in this study are listed in Supplementary Table 6. Bacterial strains were incubated in Luria-Bertani (LB) medium (containing 10 g L$^{-1}$ tryptone, 5 g L$^{-1}$ yeast extract and 10 g L$^{-1}$ NaCl) at 37 °C for routine precultivation and cloning procedures unless stated otherwise. LB medium was supplemented with 50 μg mL$^{-1}$ 5-aminolevulinic acid (ALA) for *E. coli* ST18 propagation. All the aerobic and anaerobic growth characteristics assays were carried out in L

**Table 2 | $^{15}$N-dilution experiments with maize inbred line 93-63 and different bacterial inoculants**

|           | KoΔ*nif*          | Ko                | Ko424             |
|-----------|-------------------|-------------------|-------------------|
| NF%       | 0.0 ± 1.1[a,b]    | 1.8 ± 0.9[a,c]    | 14.1 ± 1.2[b,c]   |
| NF% (−seed) | 0.0 ± 4.2[d,e]  | 2.1 ± 2.7[d,f]    | 25.6 ± 1.4[e,f]   |

Both a and d indicate that the significance between the two groups is NS. b, c, e and f all indicate the significance between the two groups is **.

The percent of N derived from the atmosphere (%NF) was measured after 12 days inoculation. 0.5 mM $^{15}$NO$_3^-$ was added at the time of inoculation. In the NF% row, the $^{14}$N in the seed is ignored. In the NF% (−seed) row, the $^{14}$N in the seed is considered and subtracted. KoΔ*nif*, inoculation with the *nif* gene cluster deletion mutant of *K. oxytoca*; Ko, inoculation with the wild type strain of *K. oxytoca*; Ko424, inoculation with the GS-P95L mutant strain of *K. oxytoca*. The means and SEMs were calculated based on at least six biological replicates. Statistical significance is indicated as **$P ≤ 0.01$ analyzed using one-way ANOVA with Student's *t*-test. NS, non-significant. Source data are provided as a Source Data file.

medium[50] with modification, containing 7.5 mM KH$_2$PO$_4$, 17.22 mM K$_2$HPO$_4$, 3.42 mM NaCl, 2 mM MgSO$_4$, 146 μM ferric citrate, 0.1 μM CuCl$_2$, 0.1 μM ZnSO$_4$, 1 μM CaCl$_2$, 0.73 μM MnCl$_2$, 0.21 μM Na$_2$MoO$_4$, 0.6% glucose. Where specified, NH$_4$Cl was added as nitrogen source, at concentrations indicated in the text. For solid media, 1.5% agar was added. For anaerobic liquid cultivation, a vacuum pump was used to exhaust the air in the anaerobic bottle, and then high purity nitrogen was filled into the anaerobic bottle. For anaerobic solid cultivation, the petri dish was placed in a 2.5-L anaerobic jar (Thermo Fisher Scientific; AG0025A) equipped with an anaerobic gas-generating sachet (Thermo Fisher Scientific; AN0025A) and an oxygen indicator (Thermo Fisher Scientific; BR0055B). Media were supplemented with antibiotics at the following concentrations: tetracycline (Tc) 10 μg mL$^{-1}$, kanamycin (Km) 50 μg mL$^{-1}$, Ampicillin (Amp) 75 μg mL$^{-1}$ and chloramphenicol (Cm) 25 μg mL$^{-1}$. In addition, the antibiotic concentration was halved under anaerobic incubation.

### Construction of mutant strains

The *glnA* P95L point mutants of *E. coli* (Ec424) and *K. oxytoca* (Ko424) were constructed using lambda Red-mediated homologous recombination[51] using temperature-sensitive plasmid pKD46 which expresses Red recombinase. Plasmids or DNA fragments were transferred into wild-type strains through transformation, electroporation or conjugation with ST18 cells. Initially to provide a *glnA* deletion strain with an antibiotic resistance cassette, the FRT-flanked kanamycin

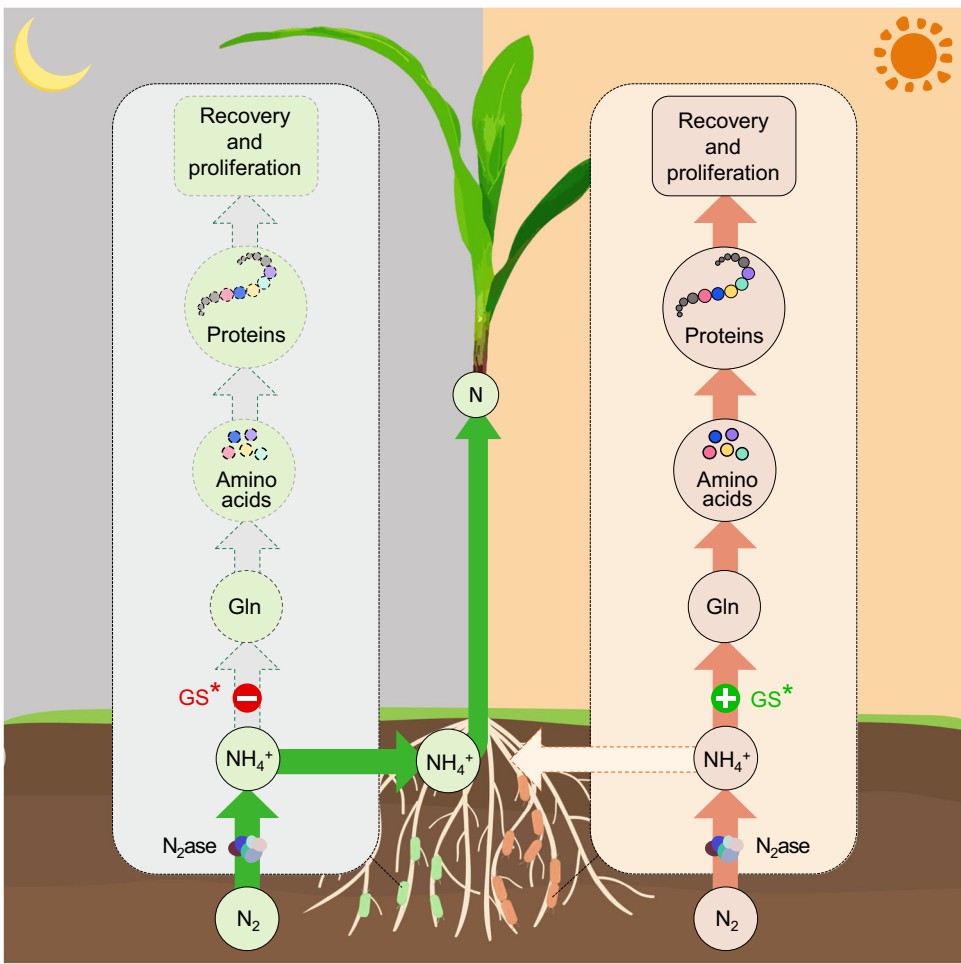

**Fig. 8 | Schematic diagram of the diurnal thermo-controlled diazotrophic lifestyle switch.** The switch uncouples biological nitrogen fixation (BNF) from ammonia assimilation in a diurnal temperature-dependent manner, enabling the bacteria to behave as free-living diazotrophs during the day to recover and proliferate (as illustrated in the right panel); and only during the night, their growth is sacrificed to enable ammonia donation to crop plants (as illustrated in the left panel). N$_2$ase, nitrogenase.

resistance (Km^R) gene with upstream and downstream homologous fragments (50 bp respectively) of *glnA* was amplified, and cloned into a plasmid pEX18Tc containing a *sacB* counter-selectable marker. The resulting plasmids were introduced into *E. coli* and *K. oxytoca* together with pKD46 respectively. The *glnA::Km* knockout mutants were obtained by resistance screening and counter selection with 10% sucrose. Then *glnA* homologous fragments containing the P95L single point mutation (CCT to CTT for *E. coli*, and CCA to CTG for *K. oxytoca*) were transferred into the *glnA::Km* knockout mutants. Ec424 and Ko424 mutants were obtained in L medium containing 15 mM NH$_4$Cl (but without glutamine), and after PCR amplification the DNA sequence was confirmed.

The *glnE* deletion mutant of Ko424 (Ko424Δ*glnE*) and *glnA* and *glnE* double-deletion mutant of BL21(DE3) (BL21(DE3)Δ*glnA*Δ*glnE*) were also constructed by lambda Red-mediated homologous recombination as stated above. The difference was that a FLP expression plasmid was transferred into *glnA::Km* or *glnE::Km* knockout mutant to eliminate the resistance cassette from the genome[51].

The *K. oxytoca nif* gene cluster deletion strain KoΔ*nif* was constructed with the FnCRISPR-Cpf1 system[52]. The *nif* gene cluster was targeted with crRNA-1 (ACCTCGGTGTCAAACACCAGAATA) and crRNA-2 (ACAGCTGGTGGTGATGAACGAACA) located at the C-terminal coding sequence of *nifJ* and *nifQ* respectively. Genomic sequences located 500 bp upstream of the PAM site for crRNA-1 and 500 bp downstream of the PAM site for crRNA-2 were amplified and fused together with overlapping PCR and used as a repair fragment.

The GFP-expressing *K. oxytoca* strain was constructed by electroporation with plasmid pHC60 which is a GFP expression vector.

The rifampicin-resistant derivative of *Escherichia coli* strain NCM3722 (Ec_rif^R) was isolated by selection for antibiotic resistance. NCM3722 was precultivated in LB medium overnight and ~5 × 10$^9$ cells were then collected and spread on LB plates containing 50 ng mL$^{-1}$ rifampicin. Rifampicin-resistant derivatives were screened after 12–16 h cultivation.

## Construction of pKU7824

A 26 Kb DNA fragment from the *K. oxytoca* genome containing the entire *nif* gene cluster was cloned into plasmid pCC1BAC using the CATCH method[53]. The resulting plasmid is designated as pKU7824.

## Calculation of doubling times

Bacterial strains were pre-cultivated in LB medium overnight. Cells were then washed at least three times and resuspended in L medium without nitrogen source prior to inoculation into L medium at an initial optical density (OD$_{600}$) of 0.05. For aerobic cultivation, growth curves were plotted in 24-well plates using a plate reader (BioTek Synergy neo multi-detection microplate reader) in 1 mL L medium supplemented with 15 mM NH$_4^+$. For anaerobic cultivation, strains were grown in L medium without nitrogen source in anaerobic bottles filled with N$_2$ and the OD$_{600}$ was measured every hour starting from the log phase until entry into stationary phase. Doubling time were determined during logarithmic growth using Eq. 1.

$$DT(min) = \ln(2)/[(\ln OD_B - \ln OD_A)/(t_B - t_A)] \qquad (1)$$

where DT represents doubling time, OD$_B$ represents the OD$_{600}$ value at the end of the log phase and OD$_A$ represents the OD$_{600}$ value at the beginning of the log phase. $t_B$ and $t_A$ represents the times (in minutes) that correspond to the OD$_{600}$ values for OD$_B$ and OD$_A$ respectively.

## Measurement of glutamate and glutamine

Cells were incubated in L medium under aerobic or anaerobic conditions at different temperatures, as described above. Bacterial samples were taken during mid-log phase. 10-mL cultures were rapidly harvested in chilled tubes. Then the cell pellets were immediately frozen in liquid nitrogen and stored at −80 °C prior to assay. Intracellular glutamate and glutamine concentrations were quantified by using Glutamine/Glutamate-Glo™ Assay (Promega; J8021). Briefly, 0.3 M HCl was added to obtain the cell lysates and then the same volume of 450 mM Tris (pH 8.0) was added to neutralize the solution. After centrifugation, the supernatants were transferred to new tubes. Samples were diluted with PBS to suitable concentrations and pipetted into a 96-well plate for the bioluminescent assay according to the manufacturer's instructions. Luminescence was recorded by using a plate reader (BioTek Synergy neo multi-detection microplate reader). To measure the dry weight, 500-mL mid-log phase cultures were collected at the same time as the above-mentioned cell harvesting, and washed in physiological saline solutions. Then cell pellets were dried to constant weight at 80 °C in a thermoelectric incubator. A value of 2 was used to convert glutamine and glutamate concentration from nmol mg$^{-1}$ dry weight to mM[30].

## β-galactosidase assay

The *E. coli glnA* promoter reporter plasmid pKU101[54], carrying the P*glnA::lacZYA* fusion, and the *K. oxytoca nifLA* promoter reporter plasmid pKU805, carrying the P*nifLA::lacZYA* fusion, were both derived from pGD926[55]. In order to amplify the *nifLA* promoter and its upstream sequence, two primers were synthesized: 5′-CCC<u>AAGCTT</u>-GAAAATACGCGTTCTC-3′ (p1, *Hind*III) and 5′-CG<u>GGATCC</u>AGCATCA-TATTCAGGGTC-3′ (p2, *Bam*HI). Restriction sites present in oligonucleotide primers used for cloning are underlined. The DNA fragment restricted by *Hind*III and *Bam*HI was inserted into pGD926 to produce plasmid pKU805. pKU101 and pKU805 were subsequently transformed into Ec and Ec424 to generate the promoter reporter strains. Cells carrying pKU101 were incubated either aerobically at 37 °C or anaerobically at 30 °C in L medium supplemented with 15 mM NH$_4^+$. Cells carrying pKU805 were incubated anaerobically at 30 °C in L medium supplemented with increasing NH$_4^+$ concentrations. 5 mL cultures were harvested during mid-log phase, centrifuged at 1844 × *g* for 3 min and stored at −80 °C prior to analysis. β-Galactosidase activity assays were performed according to Miller and Stadtman[56]. Upon thawing for measurement, cells were washed with 0.9% NaCl solution and resuspended to determine the OD$_{600}$. 500 μL samples diluted with 0.9% NaCl to suitable concentrations were added to 500 μL Z buffer[56], and permeabilized by adding 20 μL trichloromethane and 20 μL 0.1% SDS, vortex for 10 s, and holding at 28 °C for at least 5 min. The reaction was initiated by adding 200 μL ONPG and incubated at 28 °C. 500 μL 1 M Na$_2$CO$_3$ was added to stop the reaction when visible yellow color has developed. The OD$_{420}$ and OD$_{550}$ of supernatants were determined, and the Miller units were calculated by using the equation according to Miller and Stadtman[56].

## Quantitation of ammonia

Cells were incubated in L medium under anaerobic conditions, as described above. 500 μL of the culture was filtered through a 0.22-μm-pore-size filter and stored at −20 °C prior to assay. Ammonia from culture supernatants was measured by using Ammonia Assay Kit (Sigma-Aldrich; AA0100) according to the manufacturer's instructions.

## Nitrogenase activity assay

To measure the nitrogenase activity of *E. coli* and *K. oxytoca* derivates, the C$_2$H$_2$ reduction assay[57] was used. Cells were incubated in L medium under anaerobic conditions at different temperatures, as described above. 3 mL culture, which was taken during mid-log phase, was rapidly transferred to 30 mL sealed tubes. Air in the tubes was repeatedly evacuated and replaced with argon and 3 mL C$_2$H$_2$ was then immediately injected into the tube. After incubation at the indicated temperature for ~4–10 h, the gas phase was analyzed with a Shimadzu GC-2014 gas chromatograph. Data presented are mean values based on at least three biological replicates.

## NMR analysis

Strains were cultured anaerobically in N-free L medium. The headspace of the anaerobic flask was displaced with either $^{15}N_2$ gas, or $^{14}N_2$ gas, or argon. After incubation for 72 hr at 23 °C, the supernatant was separated, filtered and frozen at −20 °C for further analysis. Samples were quantified for ammonia as described for the ammonia quantification assay and prepared for $^1$H-NMR analysis by addition of 50 µL deuterated dimethyl sulfoxide (DMSO) and 25 µL concentrated HCl to 425 µL samples. Standards were prepared equivalently, using 3 mM $^{14}NH_4Cl$ and 3 mM $^{15}NH_4Cl$ (Aladdin, 99% atom) in uninoculated buffer medium. Spectra were collected using a 600 MHz Bruker Avance NMR instrument equipped with a room temperature probe for ammonia[27,38].

## Co-culture of *K. oxytoca* and Chlorella sorokiniana

*K. oxytoca* strains were pre-cultivated in LB medium overnight. *Chlorella* was pre-cultivated in BG11 medium (Psaitong; catalog number B11977) in an illuminated incubator at 28 °C for 5 days. After twice washing in L medium to remove exogenous nitrogen, *K. oxytoca* strains were streaked in a fan-like arrangement on L medium agar plates with *Chlorella* alternating between *K. oxytoca* strains. Plates were incubated at the indicated temperatures in transparent anaerobic jars equipped with an anaerobic gas-generating sachet and grown under light illumination for 7 days.

## Quantification of chlorophyll *a* content in *Chlorella*

*Chlorella* was collected in a centrifuge tube, and after measurement of fresh weight it was diluted to an appropriate concentration with PBS buffer. 3 mL of absolute methanol was added to 1 mL of *Chlorella* suspension and was mixed by vortex. After resting in the dark for 15 min, the cells were centrifuged ($10,625 \times g$ for 5 min) and the supernatant was removed and transferred to a new centrifuge tube. The absorbance of the supernatant at 652 nm and 665 nm was measured with a spectrophotometer and recorded as A652 and A655, respectively. If the absorbance value was greater than 1, the supernatant was diluted and measured again. The chlorophyll *a* content was determined according to Eq. 2.

chlorophyll *a* (mg per g fresh weight)
$$= (-8.0926 \times A652 + 16.5169 \times A665)/\text{fresh weight (mg) per mL of } Chlorella \text{ liquid} \quad (2)$$

## Maize culture and medium

Maize (*Zea mays*) was grown in a hydroponic system. Plants were cultured in the greenhouse with photoperiod of 12 h between light and dark intervals. The incubation temperature was 30 °C in the light (7 am to 7 pm) and 23 °C in the dark (7 pm to 7 am) with humidity maintained at ~60–70%. Maize was cultured in Hoagland nutrient solution, which consists of 0.5 mM $KH_2PO_4$, 0.75 mM $KH_2PO_4$, 0.65 mM $MgSO_4 \cdot 7H_2O$, 0.1 mM KCl, 1 µM $MnSO_4 \cdot H_2O$, 1 µM $ZnSO_4 \cdot 7H_2O$, 0.1 µM $CuSO_4 \cdot 5H_2O$, 0.02 µM $Na_2MoO_4 \cdot 2H_2O$, 1 µM $H_3BO_3$, 0.1 mM $FeSO_4 \cdot 7H_2O$ and 0.1 mM EDTA-$Na_2$[58,59] with the pH adjusted to 6.2. No additional carbon source was added during all cultivations of maize.

## Maize inoculation and growth measurement

Surface disinfection of maize seeds was carried out with 10% $H_2O_2$ for 30 min at 28 °C, followed by washing 4–5 times with sterilized distilled water. Seeds were then soaked in saturated $CaSO_4$ overnight and washed 6–7 times sterilized distilled water. Germination was carried out in vermiculite, at about 0.5 cm depth with an appropriate amount of distilled water, incubated in the greenhouse for 6 days. When the maize seedlings reached the stage of one leaf, healthy seedlings were selected and the vermiculite attached to the roots was removed by washing, prior to incubation in the hydroponic system. The endosperm of the maize seedlings was removed to restrict carry over of storage nutrients (especially nitrogen sources). Bacterial inoculation was carried out by incubating the whole root system of the maize for 1 h with $10^9$ bacterial cells per mL. Note: the bacterial cells were prepared as follows: the bacteria were grown in LB medium to an $OD_{600}$ of 0.6–0.8 in the log phase. Subsequently, the bacterial cells were harvested by centrifugation ($1844 \times g$ for 3 min), washed and resuspended in corresponding plant nutrient solution to the required cell density ($10^9$). Thin sponges were wound around the junction of roots and stems to serve as support for the seedlings. Seedlings treated in the same way were cultured in the same hydroponic system as a group. Maize plants were grown for 9 days, prior to harvesting for analysis of dry weight and N content respectively. Six plants were grown per pot filled with 3.5 L nutrient solution. These experiments were repeated at least three times.

To determine plant dry weight, plants were placed into kraft paper bags, and dried in an oven at 80 °C until the weight no longer changes. To measure nitrogen content, plant tissues were ground into a powder with a mortar and pestle, and analyzed for total nitrogen using an elemental analyzer (vario MACRO cube CHNOS Elemental Analyzer, Elementar Analysensysteme GmbH, Hanau, Germany). After bacteria inoculation, maize plants were grown in the hydroponic system supplemented with 0.5 mM 30% $^{15}$N-labeled calcium nitrate for 12 days, prior to harvesting for analysis of the % of N derived from the atmosphere (%NF). %NF was calculated using Eq. 3.

$$\%NF = (1 - A/B) \times 100 \quad (3)$$

where A is the atom %$^{15}$N excess in the nitrogen-fixing system and B is the atom %$^{15}$N excess in the reference plant (Ko$\Delta nif$)[60].

## Pheophytin analysis

Healthy maize seedlings reached the stage of one leaf were selected, inoculated with bacteria and incubated in the hydroponic system as described in the previous section. The hydroponic system was placed in clear and gas-tight plastic bags, inside which 50% of the gas space was displaced with $^{15}N_2$ gas and the final oxygen concentration in the gas mixture was 20%. 1% $CO_2$ was added daily to keep the $CO_2$ indicator yellow[61]. After 8 days planting, chlorophyll of the third young leaves was extracted and converted to pheophytin by concentrated hydrochloric acid[27,62]. Pheophytin $^{15}$N isotope abundances were analyzed by the LTQ Orbitrap XL Basic System (Orbitrap XL, Thermo Fisher), operating in electrospray, positive-ion mode. Each sample was injected into the mass spectrometer at a flow rate of 10 µL min$^{-1}$ for 30 s, with a spray voltage of 5000 V, a capillary temperature of 275 °C, a capillary voltage of 45 V, a tube lens of 95 V. Pheophytin isotope abundances were analyzed under these conditions, run time for 30 s and collection range was 500 - 900 $mz^{-1}$. Standards (Chlorophyll *a*, Sigma-Aldrich) were prepared following the same pheophytin conversion method. The fractional $^{15}$N isotope enrichment of the uninoculated control was subtracted from values derived from inoculated plants.

## Reporting summary

Further information on research design is available in the Nature Portfolio Reporting Summary linked to this article.

# Data availability

A reporting summary for this article is available as a Supplementary Information file. Data supporting the findings of this work are available within the paper and its Supplementary Information files. Primers used for quantitative RT-PCR are listed in Supplementary Table 7. Source data are provided with this paper.

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

## Acknowledgements

We thank Prof. Jinsheng Lai (China Agricultural University) for providing us with maize inbred lines and Prof. Chengcai Chu (Institute of Genetics and Developmental Biology, Chinese Academy of Sciences) for provision of Japonica seeds. We thank Mr. Shaohan Wu for carrying out some of the very initial stage of the work. We thank the NMR facility of the National Center for Protein Sciences at Peking University for assistance with nitrogen isotope analysis and Dr. Xiaogang Niu for help with sample testing and data acquisition. The measurements of mass spectrometry were performed at the Analytical Instrumentation Center of Peking University, we thank the Analytical Instrumentation Center in Peking University for assistance with the measurements of mass spectrometry. We also thank Dr. Wei Huang (The Monsanto Company) for providing us detailed seasonal temperature profiles for cereal plantation in central China. This research was supported by the National Science Foundation of China (research grant 32020103002 to Y.-P.W.), the National Key R&D Program of China (research grant 2019YFA0904700 to Y.-P.W.), the SLS-Qidong Innovation Fund, and the State Key Laboratory of Protein and Plant Gene Research (research grant B02 to Y.-P.W.). Y.-P.W. is a recipient of a grant from the National Science Fund for Distinguished Young Scholars (grant 39925017). D.Q. was supported by the National Science Foundation of China (research grant 32100196). R.D. was funded by the UK Research and Innovation Biotechnology and Biological Sciences Research Council (research grant BBS/E/J/000PR9797 and the Royal Society (International Collaboration Award ICA\R1\180088).

## Author contributions

D.Q., Z.-X.T., R.D., and Y.-P.W. designed research; Y.T., D.Q., Z.-X.T., W.C., Y.M., J.W., J.Y., and D.Y. performed research; Y.T., D.Q., R.D., and Y.-P.W. analyzed data; and Y.T., D.Q., R.D., and Y.-P.W. wrote the paper.

## Competing interests

The authors declare no competing interests.
