## [Peer Review File · Nature Communications]

Diurnal switches in diazotrophic lifestyle increase nitrogen contribution to cerealsReviewers' Comments:

Reviewer #1:

Remarks to the Author:

This is a valuable manuscript that explores the effects of temperature variation on nitrogen fixation by *Klebsiella oxytoca*. Temperature sensitive mutants have been known for a long time and surprisingly a *Klebsiella oxytoca* mutant in GS turned out to be temperature sensitive and could be used in crops that face daily temperature changes. Notably, best plant effects are found with diurnal temperature variation.

Best results were obtained in maize in comparison to rice. It is known that maize produce large amounts of exudates. Day and night root exudation may be different and it may have effects on bacterial nitrogen fixation as well as on the microbial density on roots that may condition the competition with roots for available ammonium. Furthermore plants may absorb ammonium differently at day or night. These questions may be addressed in the future using this work as a basis.

The paper may be organized to have all co-cultures together before effects on plants.

The large introduction on gene regulation and functions (lines 85-116) may be summarized specially the part in the presence of nitrogen. So many details and figures seem not needed to further follow and understand the paper. On the contrary more information on the effects of different bacteria that excrete nitrogen should be provided (lines 117-121).

Reviewer #2:

Remarks to the Author:

This paper is an interesting paper which builds on extensive literature dating back to the 1990s that identified glutamine synthetase mutants including the temperature sensitive P95L point mutant. The literature is reviewed extensively and provides a good background for the current paper. The paper is very well written, requiring no editorial editing. I have a few comments and, in spite of its overall high quality cannot recommend it for publication at this time.

Figure 1: lines 170-186; in these *E. coli* experiments incorporating the P95L mutant *glnA* experiments were carried out at temperatures of 37C vs 30C. I was not clear how these temperatures affected the P95L mutant glutamine synthetase – this needs to be explained since all of the description of the mutant focused on temperatures of 23C vs 30C.

Figure 3GH – this is a visually convincing figure but would be improved if there was also some way to quantify the result.

Figure 4 B/C D/E. These data demonstrate the effect of Ko424 (the *Klebsiella* harboring P95L *glnA*). The results are suggestive that Ko424 may be delivering reduced to the plant (as illustrated in Fig 4I but does not prove it. In my opinion, these type of results need to be corroborated with some measurement of Nitrogen Derived from Atmosphere (Ndfa). This typically measures $^{14}\text{N}/^{15}\text{N}$ ratios with the soil or liquid culture supplemented with ^{15}N .

Figure 2: similarly the NH_4^+ excretion data would have been more informative if the cultures had been supplemented with $^{15}\text{N}_2$ gas to demonstrate the NH_4^+ excretion was derived from nitrogen fixation.

Reviewer #3:

Remarks to the Author:

Engineering plant associated free-living diazotrophs to release ammonium in close proximity to the plant roots is probably the more direct biological approach to start replacing chemical fertilizers in

cereal crops. As the authors point out, it is first necessary to uncouple biological nitrogen fixation from nitrogen assimilation in the bacterial donor.

There has been a somewhat futile race to show which mutations and culturing conditions could yield higher amounts of excreted ammonium, on the expectation that higher ammonium would mean more N transfer and render better plant growth promotion. However, after many efforts in this direction, several caveats became evident. First, very high ammonium excretion is detrimental, not only to the bacterium donor but, in some cases, also to the recipient plant. Second, that after a short period of time and in the absence of an efficient sink ammonium was taken up again by the donor bacterium, thus creating a short-lived metabolic benefit. Third, and perhaps most important, that genotypes were frequently unstable and revertants appeared.

In my opinion, the work described in this manuscript represents is a new paradigm. It is a breakthrough. The authors find, characterized, and exploit a 'natural' metabolic (and genetic) oscillator. The word 'metabolic' should go first because the effect originates by oscillating flux of glutamine, a key metabolite that is also widely used to sense C/N balance. Then, of course, glutamine oscillations affect expression of an array of genes (elegantly shown in this work). It is not easy for synthetic biologists to engineer a good oscillator. However, the oscillator characterized here depends on a single amino acid change and it is surprisingly robust. The discussion should compare this oscillator with some of those engineered by synthetic biologists. Further advantages are its genetic stability and the fact that it responds to temperature oscillations, an important and consistent parameter in the field (perhaps only second to the circadian clock).

The effect of P95L glutamine synthetase mutation is characterized in much detail at 'low' and 'high' temperature. Biochemical studies of purified GS, in vivo levels of the most relevant metabolites (glutamine, glutamate, ammonium), transcription of genes which expression should be altered in GS mutants, ammonium excretion and donation to green algae or another bacterium, etc. Importantly, the effects are consistent in the original nitrogen-fixing host, *Klebsiella oxytoca*, and in an engineered nitrogen-fixing *E. coli* strain.

For me one surprising observation was the lack of revertants or suppressors. Especially because the phenotype was due to a single amino acid mutation and a because the affected activity is at the junction of carbon and nitrogen assimilation. However, the data shown in Figure S6 is convincing. This is probably due to the oscillating nature of the phenotype. In this context it is important to note that 20 generations take a long time in the environment, so the ammonium excretion phenotype will likely also very stable in the wild.

The analysis of P95L GS on growth promotion and nitrogen enrichment of japonica rice and nine inbred maize lines is also very detailed. The effects are not very pronounced, especially if we consider that the analyzed plants are very young and the nitrogen demand at that age is not very high. Nevertheless, the conclusions regarding growth promotion are restrained (490-493 and 508-510) and therefore fully justified by the results.

Minor comments:

- Ammonium consumption in Figure 2C and 2E is very similar. Are those differences statistically significant?
- Figure S9 refers to maize hydroponic cultures. Although it is easy to imagine that this profile is the same for rice, referring to figure S9B here may confuse the reader. Perhaps it is better to extract panel B from S9 and place it in a separate supplemental figure.
- Lines 409-411. This is an important point. It should be explained in more detail here rather than just referring to Wang 2016.

Reviewer #4:
Remarks to the Author:
Overview

This paper reports a P95L mutant of glutamine synthetase (GS) that leads to temperature sensitive activity of GS. This mutant is significant because it leads to temperature dependent exudation of ammonia, allowing cells to grow while ammonia is not exuded. Overcoming the tradeoff between growth and ammonia production is a major hurdle in the field of engineering BNF. The study does an OK job at characterizing the mutant, though does not lead to any conclusion of how the mutation causes temperature-dependent activity of GS. The study then applies the mutant to plant-growth promotion experiments and shows improved growth of maize, but does not rigorously show nitrogen transfer to plants.

Overall, the paper is worth publishing (though not sure caliber is currently sufficient for nature communications) and will need to be revised through detailed comments below pertaining to introduction, results and discussion sections. Most importantly, several claims need to be made more quantitative and softened. The literature is full of plant-microbe interaction studies that claim nitrogen is provided to plants, but few papers are very careful in demonstrating that nitrogen is actively moved from the atmosphere into plant-specific molecules during the course of plant-microbe association (isotope labelling needed).

Detailed comments

This paper is mainly about a GS mutant. Therefore, the part of the introduction that pertains to regulation of nitrogen metabolism could focus more on GS / glnA. E.g. include an overview of transcriptional control of glnA. Also, authors might want to mention what different purposes posttranslational modulation of GS activity serves compared to transcriptional regulation of glnA? (e.g. one of these is faster and primarily used to respond to rapidly changing N conditions). Finally, GS is not present in Figure 1A, and would be helpful to include, to illustrate how regulation is relevant to GS.

Sentence starting at line 112 ("Inhibiting...") needs to be rewritten to connect better to the material above it. Also, causal words "therefore" and "since" are not supported by the information in this sentence. Something along the lines of: "If the above-described nitrogen regulatory system is perturbed such that ammonia is assimilated slower by GS than it is synthesized by nitrogenase, it will accumulate extracellularly because..." would be a better start.

Starting at line 116, a more detailed overview of previous work pertaining to decreasing GS activity and yielding ammonia excretion would be useful to place this work into context. Consider briefly describing by what mechanisms GS activity has been lowered in previous work (including repression of glnA transcription, PTM deactivations via uATs, different point mutations in glnA...)

Adjust lines 149 and 341 to clarify why "seasonal temperature profile" is mentioned, followed by details on day/night profile. More importantly, are these air temperatures? How do the temperatures of soil and water change in the target environment? How relevant are the temperature data in Figure S9B to these real-world conditions? (This might be better moved to the discussion section).

Last paragraph of the introduction: please briefly include under what conditions plant experiments demonstrated results. The scope of the work is greatly impacted by whether these experiments were hydroponic mono-association studies vs non-sterile soil or even field trials. Also specify what key data were collected: isotope labelling N incorporation data vs dry weight measurements are a big difference in scope of the study and would be good to specify here. Finally and importantly, these modifications

should also be made to the abstract.

The start of the results section might be a good place to remind readers that *glnA* encodes GS (since this is a broad audience journal).

Second paragraph of results: it would be useful to include more context around why *glnHp2* and *nac* promoters are examined (they are previously not introduced).

Second paragraph of results and in general: writing would be improved by including numerical data in the text pertaining to qRT-PCR results. This makes the text more quantitative and accurate than solely relying on words e.g. "strongly activated". Importantly, this issue is present throughout the manuscript and can be improved in several results paragraphs.

Figure 2A could be improved if pKU7824 inside the bacterium schematic were replaced with a functional description like "nif gene cluster" (the name of the plasmid is already written above the schematic).

Line 240: "is limited by..." is strong. Consider "might be limited by..." as other complex regulation could be at place.

Sentence starting at line 241 until end of paragraph: The evidence presented that *Ec* pKU7824 consumes more ammonia from the media compared to *Ec424* pKU7824 does not directly lead to the conclusion that the difference must have been made up by nitrogen fixation by *Ec424*. For example, differences in growth rate or bacterial N content could explain this. Therefore, these sentences should be softened and alternative causes addressed.

Line 262: wouldn't these data suggest that growth defects are more precisely attributed to slower nitrogen assimilation through P95L at 23C compared to 30C rather than nitrogen fixation?

Major question (related to Figures 2F and 3F): Did the authors collect time courses of ammonia accumulation in the media? How were the rates of accumulation affected by temperature in the *Ec424* pKU7824 strain? It is important to be careful in determining the maximum ammonia excretion, as it is known in the literature that ammonia excreting microbes are genetically unstable and will revert to assimilating previously excreted ammonia in culture. Thus, peak ammonia levels will be reached at different times depending on the strain and condition of the experiment (authors allude to this in Figure captions, but no time course data are to be found anywhere in MS or SI – please include). The study in Figure S6 is a good start to investigate stability, but more should be done. How were 20 generations selected, and does stability remain beyond 20 generations? Are mutants present after pant-coculture and several day-night cycles?

Paragraph about biochemical properties of the P95L variant could be improved. The paper is all about characterizing this mutant, and the burning question on the reader's mind is "how does P95L lead to temperature dependent deactivation of GS?" Can the authors venture a few hypotheses? E.g. it was confirmed that adenylation by *glnE* is not the culprit (good), but what about a structural analysis of where the mutation falls? E.g. is it near the active site, near allosteric pockets, on the faces that interact to form the heterododecameric GS? As part of this analysis, it would be helpful to include a PyMOL or otherwise generated 3D structure that shows the mutation in Figure 1.

Paragraph starting at line 329: include in text that this experiment was run anaerobically to allow for nitrogen fixation.

Paragraph starting line 351: include numeric growth promotion data in text please. Also include in text a few more details about the experiment, especially that it was in mono-association. How and in what quantity were microbes added to pants? These details are very important to interpreting the data and

should not be left to the Methods section. Also, looking at the data in Figure S7, growth was not promoted enough to warrant use of the word "significantly" (line 356). If actual numbers are included in the text, such phrases can be interpreted more quantitatively.

Question about how plant line 96-63 was selected: What do the authors mean by "responded positively" (line 368) – please be quantitative and precise. Also, was this initial screen of 8 lines repeated in an independent experiment to yield the same results? Finally, can the authors postulate and/or confirm any reason for why 96-63 stood out? (e.g. more root exudates?)

For the rice experiments (line 355), the authors state no carbon source was added. However, for the maize experiments, this is not explicitly stated (line 377). Since the presence of carbon is a major factor in microbial proliferation, the text would be improved by being explicit about its presence or absence and not have the reader page through SI and Methods.

Have the authors repeated the plant growth promotion experiments in an independent iteration? Given the high degree of complexity in these types of experiment, showing data of independent iterations would improve the work significantly.

None of the plant experiments have shown that nitrogen moved from the air into the plant via active nitrogen fixation and exudation through Ko424 during the co-culture of plants and microbes, as for that, isotope labeling studies are needed. Therefore, all claims that state nitrogen is contributed or provided to plants need to be softened (e.g. line 391). Including experiments showing direct incorporation (not dilution studies) of ^{15}N from gas to plant-specific molecules (e.g. pheophytin) would significantly elevate this study.

Have the authors tested plant growth promotion experiments in soil and with presence of other microbes?

Minor suggestion: the bacterial co-culture experiment section of the Results might be better integrated ahead of the plant experiments. Additionally, it might be better to call the section "quantitative estimate" rather than "analysis".

Line 446: Specify that co-culture is bacterial.

Line 447: Not sure the limited scope of experiments around stability warrant the phrase "incredibly stable".

The discussion section could be improved by considering differences between real-world application conditions and experimental conditions of this study (e.g. presence of other microbe / competition, soil vs hydroponics, different temperature profiles as a function of depth).

Point-by-point response to the reviewers' comments

We thank the reviewers for their constructive comments on our manuscript. Our point-by-point answers are presented in italicised blue text

Reviewer #1 (Remarks to the Author):

This is a valuable manuscript that explores the effects of temperature variation on nitrogen fixation by *Klebsiella oxytoca*. Temperature sensitive mutants have been known for a long time and surprisingly a *Klebsiella oxytoca* mutant in GS turned out to be temperature sensitive and could be used in crops that face daily temperature changes. Notably, best plant effects are found with diurnal temperature variation.

We appreciate the reviewer's recognition of our work. We also appreciate this reviewer's constructive comments and suggestions.

Best results were obtained in maize in comparison to rice. It is known that maize produce large amounts of exudates. Day and night root exudation may be different and it may have effects on bacterial nitrogen fixation as well as on the microbial density on roots that may condition the competition with roots for available ammonium. Furthermore plants may absorb ammonium differently at day or night. These questions may be addressed in the future using this work as a basis.

We agree that there may be differences in ammonium absorption by plants during the diurnal cycle and that the composition of root exudates may also differ during the light-dark cycle. However, as suggested by the reviewer, these are points that require addressing in the future and further experiments to examine these fluctuations are outside the scope of this manuscript.

The paper may be organized to have all co-cultures together before effects on plants.

These is a very logical suggestion. We have now rearranged the manuscript so that the section on ammonium donation to non-diazotrophic microorganisms appears before the plant experiments.

The large introduction on gene regulation and functions (lines 85-116) may be summarized specially the part in the presence of nitrogen. So many details and figures seem not needed to further follow and understand the paper.

We have removed some of the details of regulation in the introduction to focus more on the regulation of GS, as also requested by Reviewer #4

On the contrary more information on the effects of different bacteria that excrete nitrogen should be provided (lines 117-121).

We have expanded this section of the introduction to include more details of bacteria that have been engineered for ammonia excretion and the influence of these variant strains on plant growth. We have focused this entirely on manipulations that influence GS activity to avoid bloating the text of the introduction.

Reviewer #2 (Remarks to the Author):

This paper is an interesting paper which builds on extensive literature dating back to the 1990s that identified glutamine synthetase mutants including the temperature sensitive P95L point mutant. The literature is reviewed extensively and provides a good background for the current paper. The paper is very well written, requiring no editorial editing. I have a few comments and, in spite of its overall high quality cannot recommend it for publication at this time.

Figure 1: lines 170-186; in these *E. coli* experiments incorporating the P95L mutant *glnA* experiments were carried out at temperatures of 37C vs 30C. I was not clear how these temperatures affected the P95L mutant glutamine synthetase – this needs to be explained since all of the description of the mutant focused on temperatures of 23C vs 30C.

Initially, we carried out experiments with the E. coli GS-P95L mutant at 37C to make comparisons with previous studies in S. typhimurium and ensure that the phenotype was similar. However, the K. oxytoca nitrogen fixation system is temperature sensitive, so in order to determine the influence of the GS-P95L substitution on nitrogenase activity and ammonium excretion, we performed most of the subsequent experiments at 30C or below. This is now explained in the text. Although we have not examined the kinetic properties of the E. coli GS-P95L enzyme in response to temperature, the amino acid sequences of E. coli and K. oxytoca GS are highly conserved (see Supplementary Fig. 8). As described in the manuscript, we purified K. oxytoca native GS and the GS-P95L variant and examined their kinetic properties in response to temperature in the range between 20-37C (Supplementary Fig. 7). As expected, the catalytic activity of GS-P95L is lower than that of the wild type at all temperatures, but for GS-P95L, the Km for both glutamate and ammonia decreases in proportion to increases in temperature (Supplementary Fig. 7 panel E) thus enabling synthesis of more glutamine at the higher temperatures. This results in the restoration of negative feedback regulation to nitrogen regulated genes and enzymes at temperatures of 30C and above.

Figure 3GH – this is a visually convincing figure but would be improved if there was also some way to quantify the result.

This point now refers to revised Fig. 4. We have now added an additional panel to this figure in which we have measured the chlorophyll a content of the Chlorella adjacent to Ko424 in the solid medium and demonstrated that this is significantly higher at 23 compared with 30C (~ 15 fold). Please see panel 4C

Figure 4 B/C D/E. These data demonstrate the effect of Ko424 (the *Klebsiella* harboring P95L *glnA*). The results are suggestive that Ko424 may be delivering reduced to the plant (as illustrated in Fig 4I but does not prove it. In my opinion, these type of results need to be corroborated with some measurement of Nitrogen Derived from Atmosphere (NDFA). This typically measures 14N/15N ratios with the soil or liquid culture supplemented with 15N.

This point now refers to the data in revised Fig. 5. In response to this comment, we have now performed 15N dilution experiments with maize and added a new figure (Supplementary Fig.

16) to show that Ko424 provides an Ndfa (referred to in the manuscript as %NF) of ~14% when compared with reference plants inoculated with the non-nitrogen fixing strain KoΔnif and ~26% Ndfa when the ¹⁴N in the seed is considered and subtracted. Please also note that in response to reviewer #4, we have also included data using ¹⁵N₂ gas to demonstrate that fixed nitrogen derived from nitrogen fixation is incorporated into a plant-specific molecule (pheophytin) (please see Figs S17 and S18).

Figure 2: similarly the NH₄⁺ excretion data would have been more informative if the cultures had been supplemented with ¹⁵N₂ gas to demonstrate the NH₄⁺ excretion was derived from nitrogen fixation.

We agree it is important to demonstrate that the ammonium excreted by the bacteria is derived from nitrogenase activity rather than, for example, intracellular nitrogen turnover. We have therefore used proton NMR and a labelling approach to demonstrate that ammonium in the culture supernatant is derived from ¹⁵N₂ gas and therefore is a consequence of nitrogen fixation (please see new Supplementary Fig. 5).

Reviewer #3 (Remarks to the Author):

Engineering plant associated free-living diazotrophs to release ammonium in close proximity to the plant roots is probably the more direct biological approach to start replacing chemical fertilizers in cereal crops. As the authors point out, it is first necessary to uncouple biological nitrogen fixation from nitrogen assimilation in the bacterial donor.

There has been a somewhat futile race to show which mutations and culturing conditions could yield higher amounts of excreted ammonium, on the expectation that higher ammonium would mean more N transfer and render better plant growth promotion. However, after many efforts in this direction, several caveats became evident. First, very high ammonium excretion is detrimental, not only to the bacterium donor but, in some cases, also to the recipient plant. Second, that after a short period of time and in the absence of an efficient sink ammonium was taken up again by the donor bacterium, thus creating a short-lived metabolic benefit. Third, and perhaps most important, that genotypes were frequently unstable and revertants appeared.

We are very grateful for the reviewer's insightful comments regarding the potential benefits of engineering ammonium excreting strains and for clearly highlighting the caveats associated with this approach.

In my opinion, the work described in this manuscript represents a new paradigm. It is a breakthrough. The authors find, characterized, and exploit a 'natural' metabolic (and genetic) oscillator. The word 'metabolic' should go first because the effect originates by oscillating flux of glutamine, a key metabolite that is also widely used to sense C/N balance. Then, of course, glutamine oscillations affect expression of an array of genes (elegantly shown in this work). It is not easy for synthetic biologists to engineer a good oscillator. However, the oscillator characterized here depends on a single amino acid change and it is surprisingly robust. The discussion should compare this oscillator with some of those engineered by synthetic biologists. Further advantages are its genetic stability and the fact that it responds

to temperature oscillations, an important and consistent parameter in the field (perhaps only second to the circadian clock).

We are delighted that this reviewer considers that the work in this manuscript represents a new paradigm. We entirely agree with the suggestion to place more emphasis on the robust nature of our glutamine oscillator, which is dependent on a single amino acid change, in contrast to the complex manipulations often required to engineer oscillators using synthetic biology. In response to these comments, we have added more details in the Discussion to highlight comparison between our oscillator and the difficulties associated with synthetic oscillator design.

The effect of P95L glutamine synthetase mutation is characterized in much detail at 'low' and 'high' temperature. Biochemical studies of purified GS, in vivo levels of the most relevant metabolites (glutamine, glutamate, ammonium), transcription of genes which expression should be altered in GS mutants, ammonium excretion and donation to green algae or another bacterium, etc. Importantly, the effects are consistent in the original nitrogen-fixing host, *Klebsiella oxytoca*, and in an engineered nitrogen-fixing *E. coli* strain.

For me one surprising observation was the lack of revertants or suppressors. Especially because the phenotype was due to a single amino acid mutation and a because the affected activity is at the junction of carbon and nitrogen assimilation. However, the data shown in Figure S6 is convincing. This is probably due to the oscillating nature of the phenotype. In this context it is important to note that 20 generations take a long time in the environment, so the ammonium excretion phenotype will likely also very stable in the wild.

*We apologise that we did not point out in the original manuscript that two nucleotide substitutions were introduced into the *glnA* gene to encode the P95L GS variant in the *K. oxytoca* strain (Ko424). This is now clearly indicated in Supplementary Fig. 10A. Although this is likely to prevent reversion, we have not so far encountered any suppressors of the mutation. In response to reviewer #4, we performed another experiment in which we grew the bacteria for 40 generations and stability was still maintained (Supplementary Fig. 10B).*

The analysis of P95L GS on growth promotion and nitrogen enrichment of japonica rice and nine inbred maize lines is also very detailed. The effects are not very pronounced, especially if we consider that the analyzed plants are very young and the nitrogen demand at that age is not very high. Nevertheless, the conclusions regarding growth promotion are restrained (490-493 and 508-510) and therefore fully justified by the results.

Minor comments:

- Ammonium consumption in Figure 2C and 2E is very similar. Are those differences statistically significant?

The differences in Figure 2C and 2E are statistically significant and we have now included the statistics in these panels

- Figure S9 refers to maize hydroponic cultures. Although it is easy to imagine that this profile

is the same for rice, referring to figure S9B here may confuse the reader. Perhaps it is better to extract panel B from S9 and place it in a separate supplemental figure.

We have made changes as requested and extracted panel B from Figure S9 (Supplementary Fig. 14 in the revised manuscript) to a separate supplementary Figure (Supplementary Fig. 11 in the revised manuscript).

- Lines 409-411. This is an important point. It should be explained in more detail here rather than just referring to Wang 2016.

We have explained this as follows: "It has been estimated that E. coli requires about 4.5 to 5 mM of nitrogen to reach a cell density (OD₆₀₀) of 1.0 (Wang et al., 2016)".

Reviewer #4 (Remarks to the Author):

Overview

This paper reports a P95L mutant of glutamine synthetase (GS) that leads to temperature sensitive activity of GS. This mutant is significant because it leads to temperature dependent exudation of ammonia, allowing cells to grow while ammonia is not exuded. Overcoming the tradeoff between growth and ammonia production is a major hurdle in the field of engineering BNF. The study does an OK job at characterizing the mutant, though does not lead to any conclusion of how the mutation causes temperature-dependent activity of GS. The study then applies the mutant to plant-growth promotion experiments and shows improved growth of maize, but does not rigorously show nitrogen transfer to plants.

Overall, the paper is worth publishing (though not sure caliber is currently sufficient for nature communications) and will need to be revised through detailed comments below pertaining to introduction, results and discussion sections. Most importantly, several claims need to be made more quantitative and softened. The literature is full of plant-microbe interaction studies that claim nitrogen is provided to plants, but few papers are very careful in demonstrating that nitrogen is actively moved from the atmosphere into plant-specific molecules during the course of plant-microbe association.

We thank the reviewer for their detailed comments and suggestions. We agree with both reviewers # 2 and # 4 that the manuscript would be strengthened by the inclusion of experiments to show that nitrogen fixation provides fixed N to plants. Accordingly, we have included both ¹⁵N dilution and ¹⁵N₂ incorporation experiments in the manuscript to provide more definitive evidence for this.

Detailed comments

This paper is mainly about a GS mutant. Therefore, the part of the introduction that pertains to regulation of nitrogen metabolism could focus more on GS / *glnA*. E.g. include an overview of transcriptional control of *glnA*. Also, authors might want to mention what different purposes posttranslational modulation of GS activity serves compared to transcriptional

regulation of *glnA*? (e.g. one of these is faster and primarily used to respond to rapidly changing N conditions). Finally, GS is not present in Figure 1A, and would be helpful to include, to illustrate how regulation is relevant to GS.

We have now included more details of the post-translational regulation of GS activity in the introduction and alluded to the fact that PTM provided a more rapid response than transcriptional regulation. We have also included a new panel in Fig 1A to illustrate the role of GlnE in GS adenylylation

Sentence starting at line 112 (“Inhibiting...”) needs to be rewritten to connect better to the material above it. Also, causal words “therefore” and “since” are not supported by the information in this sentence. Something along the lines of: “If the above-described nitrogen regulatory system is perturbed such that ammonia is assimilated slower by GS than it is synthesized by nitrogenase, it will accumulate extracellularly because...” would be a better start.

This statement has been removed in the revised manuscript in view of the comment below and reviewer # 1’s suggestions to shorten the introduction

Starting at line 116, a more detailed overview of previous work pertaining to decreasing GS activity and yielding ammonia excretion would be useful to place this work into context. Consider briefly describing by what mechanisms GS activity has been lowered in previous work (including repression of *glnA* transcription, PTM deactivations via uATs, different point mutations in *glnA*...)

We have revised the introduction accordingly and briefly mentioned the various approaches to perturb GS activity and its expression.

Adjust lines 149 and 341 to clarify why “seasonal temperature profile” is mentioned, followed by details on day/night profile. More importantly, are these air temperatures? How do the temperatures of soil and water change in the target environment? How relevant are the temperature data in Figure S9B to these real-world conditions? (This might be better moved to the discussion section).

In the introduction we have qualified this with the sentence “This enables temperature dependent switching of the bacterial lifestyle during the day-night cycle and diurnal oscillation of ammonia excretion”.

These seasonal temperature profiles are air temperatures. Unfortunately we do not have equivalent data for either soil or water temperatures. We have added several sentences in the Discussion to acknowledge that our experiments have been performed under artificial conditions in the laboratory or greenhouse and may not be representative of real-world conditions.

Last paragraph of the introduction: please briefly include under what conditions plant experiments demonstrated results. The scope of the work is greatly impacted by whether these experiments were hydroponic mono-association studies vs non-sterile soil or even field trials. Also specify what key data were collected: isotope labelling N incorporation data vs dry weight measurements are a big difference in scope of the study and would be good to

specify here. Finally and importantly, these modifications should also be made to the abstract.

We now specify in the introduction that the plant tests were carried out under “hydroponic conditions, with a 30–42% increase in biomass and 1.4% incorporation of ¹⁵N₂ into the total N content of maize pheophytin, in the absence of added carbon source”.

We have not added all these details into the abstract as it has a limit of 150 words and is intended to be read by a broad audience.

The start of the results section might be a good place to remind readers that glnA encodes GS (since this is a broad audience journal).

We have altered the title of this section to “Characterization of a GS variant conferring altered nitrogen regulation” and stated in the first sentence that GS encodes glnA

Second paragraph of results: it would be useful to include more context around why glnHp2 and nac promoters are examined (they are previously not introduced).

We have expanded this slightly and explained that the 5 nitrogen regulated promoters were analyzed to assess the full hierarchy of Ntr-dependent gene expression.

Second paragraph of results and in general: writing would be improved by including numerical data in the text pertaining to qRT-PCR results. This makes the text more quantitative and accurate than solely relying on words e.g. “strongly activated”. Importantly, this issue is present throughout the manuscript and can be improved in several results paragraphs.

In response to this comment, we have now added details of the fold changes in each case

Figure 2A could be improved if pKU7824 inside the bacterium schematic were replaced with a functional description like “nif gene cluster” (the name of the plasmid is already written above the schematic).

We have made these changes as requested in Figure 2A.

Line 240: “is limited by...” is strong. Consider “might be limited by...” as other complex regulation could be at place.

We have changed this accordingly to “might be limited”.

Sentence starting at line 241 until end of paragraph: The evidence presented that Ec pKU7824 consumes more ammonia from the media compared to Ec424 pKU7824 does not directly lead to the conclusion that the difference must have been made up by nitrogen fixation by Ec424. For example, differences in growth rate or bacterial N content could explain this. Therefore, these sentences should be softened and alternative causes addressed.

We have softened this statement as follows: “We also measured the external ammonia during the time course of the experiment and observed that less ammonia was consumed by

the Ec424 (pKU7824) strain, potentially suggesting that this strain can fix nitrogen under nitrogen rich conditions (Fig. 2C)”

Line 262: wouldn't these data suggest that growth defects are more precisely attributed to slower nitrogen assimilation through P95L at 23C compared to 30C rather than nitrogen fixation?

When these experiments were conducted under aerobic conditions, which results in repression of nitrogen fixation, the growth defects were less substantial, implying that they may be attributed to nitrogen fixation. We have softened the conclusion to “might be associated with nitrogen fixation”

Major question (related to Figures 2F and 3F): Did the authors collect time courses of ammonia accumulation in the media? How were the rates of accumulation affected by temperature in the Ec424 pKU7824 strain? It is important to be careful in determining the maximum ammonia excretion, as it is known in the literature that ammonia excreting microbes are genetically unstable and will revert to assimilating previously excreted ammonia in culture. Thus, peak ammonia levels will be reached at different times depending on the strain and condition of the experiment (authors allude to this in Figure captions, but no time course data are to be found anywhere in MS or SI – please include). The study in Figure S6 is a good start to investigate stability, but more should be done. How were 20 generations selected, and does stability remain beyond 20 generations? Are mutants present after plant-coculture and several day-night cycles?

We have now added time courses of ammonia accumulation in the media for both the Ec424 pKU7824 and the Ko strain grown at 23C in a new figure (Supplementary Fig. 3). This shows that peak rates of ammonium accumulation occurred after maximum nitrogenase activities had been reached. Similar ammonium accumulation profiles were observed at 20C, 25C, and 27C with maximum rates occurring at different times dependent on growth rates. In all cases we did not observe apparent reassimilation of the excreted ammonium. However, this situation changed in the coculture experiments when ammonia excretion by Ko424 and ammonia absorption by E. coli occurred dynamically and simultaneously during their respective log growth phases.

We regard to stability, as mentioned in our response to reviewer #3 (who notes that 20 generations takes a long time in the environment and therefore the phenotype is likely to be stable in the wild), we repeated the coculture experiments with Chlorella in which we grew the bacteria for 40 generations and stability was still maintained (Fig 10 B).

After maize-coculture and several day-night cycles, root bacteria were collected and plated out. Sequencing the glnA gene from several colonies revealed that the mutation was still present, in accordance with retention of the ammonium excretion phenotype when the bacteria were co-cultured again with Chlorella

Paragraph about biochemical properties of the P95L variant could be improved. The paper is all about characterizing this mutant, and the burning question on the reader's mind is “how

does P95L lead to temperature dependent deactivation of GS?” Can the authors venture a few hypotheses? E.g. it was confirmed that adenylylation by *glnE* is not the culprit (good), but what about a structural analysis of where the mutation falls? E.g. is it near the active site, near allosteric pockets, on the faces that interact to form the heterododecameric GS? As part of this analysis, it would be helpful to include a PyMOL or otherwise generated 3D structure that shows the mutation in Figure 1.

We agree that a figure to depict the location of the G95L substitution in the structure of GS would be a useful addition and this has now been added (please see Supplementary Fig. 1). We have interpreted the potential for structural change induced by the substitution as follows: “The P95 residue is surface exposed in the structure of the S. typhimurium GS dodecamer (Eisenberg et al., 2000), and is not positioned within the active site formed between adjacent subunits. However, since P95 is located in a loop in the vicinity of the active site residue D65, the leucine substitution at position 95 could potentially have an impact on catalysis (Supplementary Fig. 1).”

Paragraph starting at line 329: include in text that this experiment was run anaerobically to allow for nitrogen fixation.

We have now stated that this experiment was carried out under anaerobic conditions

Paragraph starting line 351: include numeric growth promotion data in text please. Also include in text a few more details about the experiment, especially that it was in mono-association. How and in what quantity were microbes added to plants? These details are very important to interpreting the data and should not be left to the Methods section. Also, looking at the data in Figure S7, growth was not promoted enough to warrant use of the word “significantly” (line 356). If actual numbers are included in the text, such phrases can be interpreted more quantitatively.

We have now addressed these points in the text by specifying that the rice experiments were performed in coculture and have included numeric growth promotion data. We have also included details of the inoculation procedure and removed the word “significantly” in the text. (This point refers to revised Supplementary Fig. 12).

Question about how plant line 96-63 was selected: What do the authors mean by “responded positively” (line 368) – please be quantitative and precise. Also, was this initial screen of 8 lines repeated in an independent experiment to yield the same results? Finally, can the authors postulate and/or confirm any reason for why 96-63 stood out? (e.g. more root exudates?)

We have addressed this as follows: “In general, wild type K. oxytoca or the KoΔnif mutant provided little impact on maize growth compared with the uninoculated control, but three maize inbred lines responded positively to Ko424, exhibiting increases in dry weight of 14% (with B73), 20% (with Fu8701) and 29% (with 93-63) in comparison with the wild-type strain (Supplementary Fig. 13)”

We examined the potential for the Ko strain to proliferate in the root exudates from the 8 different maize inbred lines but did not observe a correlation with the observed plant growth promotion. Other factors including the ability of the plant to absorb the excreted ammonia, and the ability of the bacteria to colonize roots of different plant varieties may be important. Since the preliminary screening of the inbred lines was carried out with at least 5 independent replicates we do not consider that an additional repeat experiment is necessary.

For the rice experiments (line 355), the authors state no carbon source was added. However, for the maize experiments, this is not explicitly stated (line 377). Since the presence of carbon is a major factor in microbial proliferation, the text would be improved by being explicit about its presence or absence and not have the reader page through SI and Methods.

We have now stated that the maize experiments have been carried out without additional carbon source in the Results section as well as in the Methods

Have the authors repeated the plant growth promotion experiments in an independent iteration? Given the high degree of complexity in these types of experiment, showing data of independent iterations would improve the work significantly.

The plant growth promotion experiments have been repeated in an independent experiment (please see new Supplementary Fig. 15). This shows the same trend but with different values. As a consequence, we have averaged the biomass increase to ~30–42% and the increase in N content to ~34–38% when Ko424 is cocultured with maize grown with temperature shifts.

None of the plant experiments have shown that nitrogen moved from the air into the plant via active nitrogen fixation and exudation through Ko424 during the co-culture of plants and microbes, as for that, isotope labelling studies are needed. Therefore, all claims that state nitrogen is contributed or provided to plants need to be softened (e.g. line 391). Including experiments showing direct incorporation (not dilution studies) of ^{15}N from gas to plant-specific molecules (e.g. pheophytin) would significantly elevate this study.

We agree that the use of $^{15}\text{N}_2$ gas provides a definitive method to demonstrate that plant nitrogen is derived from nitrogen fixation. However, the necessity to enclose the plants in a $^{15}\text{N}_2$ -labelled atmosphere can impose stressful conditions. To address the reviewer's point, we have performed such studies in the absence of a bacterial carbon source to demonstrate that $^{15}\text{N}_2$ is directly incorporated into plant pheophytin in both rice and maize (please see Figs S17 and S18).

Have the authors tested plant growth promotion experiments in soil and with presence of other microbes?

We did not test plant growth promotion experiments in soil or in the presence of other microbes.

Minor suggestion: the bacterial co-culture experiment section of the Results might be better

integrated ahead of the plant experiments. Additionally, it might be better to call the section “quantitative estimate” rather than “analysis”.

We have addressed this in our response to reviewer #1, who also suggested this change. Since this section now also includes the coculture with Chlorella we have changed the title of this section to “Quantitative estimation of the influence of temperature on ammonium donation by Ko424 to non-diazotrophic microorganisms”

Line 446: Specify that co-culture is bacterial.

We have added the word “bacterial”

Line 447: Not sure the limited scope of experiments around stability warrant the phrase “incredibly stable”.

Our response to this point was listed in our comments to the reviewer above regarding stability. We consider the phrase “incredibly stable” is justified given that we repeated the coculture experiments with Chlorella for 40 generations and stability was still maintained. Furthermore, we agree with reviewer #3’s comments that “it is important to note that 20 generations take a long time in the environment, so the ammonium excretion phenotype will likely also be very stable in the wild.”

The discussion section could be improved by considering differences between real-world application conditions and experimental conditions of this study (e.g. presence of other microbe / competition, soil vs hydroponics, different temperature profiles as a function of depth).

As noted above we have added several sentences in the Discussion to acknowledge that our experiments have been performed under artificial conditions in the laboratory or greenhouse and may not be representative of real-world conditions.

Reviewers' Comments:

Reviewer #1:

Remarks to the Author:

Great revision of the manuscript

Reviewer #2:

Remarks to the Author:

The revised manuscript is greatly improved. In particular, supplementary figures 16 and 17 provide the data that was missing from the original submission. The paper reports significant and noteworthy results and, with this revision, is acceptable for publication..

Reviewer #3:

Remarks to the Author:

The authors present a revised version of the manuscript that is significantly improved. NDFA measurements using ^{15}N dilution and $^{15}\text{N}_2$ incorporation directly indicate that the *Klebsiella* mutant strain is delivering nitrogen to the plant. This has also been demonstrated by incorporation of fixed nitrogen into the plant molecule pheophytin. The authors have performed additional experiments to investigate further the phenotype of ammonium excretion to the medium and its genetic stability. These experiments are convincing.

Reviewer #4:

Remarks to the Author:

The paper is improved both in flow and content. Especially the isotope labelling studies that were requested in the previous review have the potential to significantly elevate this study compared to other studies in the BNF field.

However, I can still not recommend this study for publication at this point for several reasons, the two biggest ones being:

(1) The results from plant growth promotion and nitrogen content studies are an order of magnitude higher compared to the more rigorous ^{15}N isotope incorporation study. This difference must be discussed as it calls into question to what degree BNF via the Ko424 strain is the mechanism of action for plant growth promotion (the main hypothesis of the paper and in its title). Although there is also an isotope dilution study in this revised MS, it should be emphasized that the ^{15}N isotope integration (pheophytin) study is more indicative because it is not confounded by e.g. the difference in size of reference plant to plant of interest (size affects dilution), also because it looks at a plant specific molecule, pheophytin, and is therefore not confounded by signals coming from the nitrogen fixing bacteria that also colonize the plant, etc.

(2) As already mentioned in the initial review, the authors must fix all instances of imprecise language that is confusing nitrogen contribution / incorporation / provision with experiments that measure nitrogen content or plant growth promotion (examples pointed out in detailed comments). An increase in nitrogen content or biomass does not mean it was provided via BNF – in fact isotope incorporation studies that actually measure nitrogen contribution show that a much lower amount of N was incorporated in this study. Imprecise language such as this is unfortunately common and has not been healthy for the BNF field: together we should aim for a higher standard.

Detailed comments:

Introduction is well written!

Some parts of the MS need references: e.g. introduction paragraphs between lines 91 and 114, and also results paragraph concerning NMR study. Please check other sections that were added between original MS submission and revisions that have not been referenced.

Figure 1a caption does not match figure (top/bottom should be left/right?).

The authors might consider putting SI figure 1 (GS structure showing point mutation) into the main manuscript. It is more informative than main figures 1b-1d currently.

Concerning the point that less ammonia is consumed by Ec424 vs Ec in figures 2c and 2e and accompanying text in results: it should be pointed out in the text that this could also be due to the differences in growth rates of the two strains. The faster growing strain (Ec) is expected to consume ammonia faster. Given this more obvious explanation (compared to BNF), the authors might want to consider whether the ammonia consumption data are worthy for the main text.

Regarding stability, what happens after 72h to NH₃ accumulation in SI figure 3?

Line 321 states maximum ammonia excretion occurs at 23C. Figure 3f however shows it occurs at 20C. Please reconcile.

The NMR data (SI figure 5) is very impressive and the authors might consider showcasing it (as well as other isotope labelling data) in the main text. This type of rigorous experiment is not often done and elevates this study from others.

Section concerning "Biochemical properties of the P95L variant of glutamine synthetase" is weak because only the WT enzyme was successfully characterized. This paragraph does not add value to the manuscript unless data from SI figure 7B are better interpreted or further investigated. E.g. Line 371: please explain why the K_m of the P95L mutant could not be determined and how despite not being able to determine it, the K_m is "apparently substantially reduced". Additionally, a minor point for line 374: substantially different from what? (Clarify sentence). Unless elucidated in revisions, the section should also conclude that we do not yet understand how this mutation works biochemically.

Figure 4 panel f should come before panels d and e, as panels d and e are "sampled" from panel f; this way of presenting it is more intuitive to the reader.

Lines 391/392: nitrogen provision has not been shown in this experiment – soften conclusions.

Paragraph starting at line 393: there is a serious issue with the E. coli co-culture experiments. Constant incubation at 23C could result in a lower final OD than an oscillating temperature condition because growth rates of both strains are severely slowed at the lower temperature. This is a more likely conclusion than nitrogen provision and must be addressed. The control that is needed to strengthen the current conclusions is a 30C coculture. If this also results in a lower OD than the oscillating 23/30C condition, then the current conclusion is indeed supported. Additionally, the reader is asking themselves why this oscillation experiment was not done with algae co-culture as the section started this way – a clarifying sentence might help explain this.

Line 429: The authors should highlight that those 400 stable colonies are 100% of colonies screened. Given the previous literature reports of instability of NH₃ excretion phenotypes, this is impressive. It will also be critically viewed and a second piece of data might be needed to show stability. For example: as pointed out earlier, was ammonia reassimilated after 72 h (SI figure 3)? If not, that would

be very important to show. If yes, it calls into question the 400-colony experimental design.

Line 443: concerning the 12% increase in DW compared to all controls: SI figure 12C shows that that it is not 12% across the board but differs depending on control group. Please clarify in the text exactly what comparison was made.

Lines 481-484: authors have not shown nitrogen provision in this experiment and therefore language should be changed to reflect this (the words provision and contribution are not precise here).

Lines 536-542: concerning the discussion of the oscillator paradigm and its comparison to synthetic biology oscillators: (1) please cite examples of synthetic biology oscillators for context. (2) As a larger point, I think this section of the discussion should be reframed. The distinction between this work and other oscillators is that this work has not actually built an oscillator. The oscillator is external (temperature), whereas synthetic biology often aims to build internal oscillators – which are much more complex. Instead, this work makes use of an existing external oscillator (temperature) and exploits the activity of an enzyme mutant as well as its phenotype in response to it. Additionally, the authors might want to consider combining this part of the discussion with the current last paragraph of the discussion that talks about other potentially oscillating natural switches.

Lines 579 and 581 and 599: 34% and 18% and 11-15% respectively are measuring nitrogen content, not nitrogen contribution. Please re-word to be precise. There are no data to support that this much nitrogen is being contributed by Ko424 in this paper.

Response to the reviewers

We thank the reviewers for their constructive comments on our manuscript. Since reviewers #1-#3 did not request further amendments, we are only addressing the comments made by Reviewer #4. Our point-by-point answers are presented in italicised blue text.

Reviewer #1 (Remarks to the Author):

Great revision of the manuscript

Reviewer #2 (Remarks to the Author):

The revised manuscript is greatly improved. In particular, supplementary figures 16 and 17 provide the data that was missing from the original submission. The paper reports significant and noteworthy results and, with this revision, is acceptable for publication.

Reviewer #3 (Remarks to the Author):

The authors present a revised version of the manuscript that is significantly improved. NDFA measurements using ^{15}N dilution and $^{15}\text{N}_2$ incorporation directly indicate that the *Klebsiella* mutant strain is delivering nitrogen to the plant. This has also been demonstrated by incorporation of fixed nitrogen into the plant molecule pheophytin. The authors have performed additional experiments to investigate further the phenotype of ammonium excretion to the medium and its genetic stability. These experiments are convincing.

Reviewer #4 (Remarks to the Author):

The paper is improved both in flow and content. Especially the isotope labelling studies that were requested in the previous review have the potential to significantly elevate this study compared to other studies in the BNF field.

However, I can still not recommend this study for publication at this point for several reasons, the two biggest ones being:

(1) The results from plant growth promotion and nitrogen content studies are an order of magnitude higher compared to the more rigorous ^{15}N isotope incorporation study. This difference must be discussed as it calls into question to what degree BNF via the Ko424 strain is the mechanism of action for plant growth promotion (the main hypothesis of the paper and in its title). Although there is also an isotope dilution study in this revised MS, it should be emphasized that the ^{15}N isotope integration (pheophytin) study is more indicative because it is not confounded by e.g. the difference in size of reference plant to plant of interest (size affects dilution), also because it looks at a plant specific molecule,

pheophytin, and is therefore not confounded by signals coming from the nitrogen fixing bacteria that also colonize the plant, etc.

*While we agree that the use of $^{15}\text{N}_2$ gas provides a definitive method to demonstrate that plant nitrogen is derived from nitrogen fixation, it is well recognized in the field that it is difficult to maintain healthy plant physiology and metabolism when plants are enclosed in a sealed enclosure, where ambient levels of oxygen, CO_2 , light intensity and humidity may differ from those in the natural environment. We have performed additional experiments to further clarify this issue (Supplementary Figure 17). In our experimental setup, which involves enclosure in sealed gas-tight bags, plant biomass, chlorophyll content and the ability to assimilate ^{15}N nitrate is decreased significantly in the closed environment. Moreover, the ability of plant exudates to support the growth of *K. oxytoca* is significantly reduced (Supplementary Figure 17).*

We do not believe that our experiments are unusual in this respect. The reviewer may wish to consult the review by Philip Chalk published in 2016 (Symbiosis 69, 63-80 <http://dx.doi.org/10.1007/s13199-016-0397-8>) which provides a compilation of ^{15}N experiments carried out with various endophyte inoculants and cereal crops. It is notable that, in general, $^{15}\text{N}_2$ gas experiments carried out in closed environments result in P_{atm} % values that are an order of magnitude lower than ^{15}N dilution experiments (referred to as enrichment (E) experiments in this review, compare Table 1 with Tables 2, 3 and 5). In particular, Chalk states on page 64 that “exposure of plants to $^{15}\text{N}_2$ within an enclosure while maintaining normal plant metabolism is fraught with difficulty”.

*Some authors have circumvented this issue by including a carbon source in the growth medium. For example, Wood et al 2001 (Functional Plant Biology. 28: 969-974 <http://dx.doi.org/10.1071/PP01036>) directly compared $^{15}\text{N}_2$ transfer to wheat shoot tissue in a sealed environment in the absence and presence of added carbon. They demonstrated a 48-fold increase in newly fixed $^{15}\text{N}_2$ transfer from *Azospirillum brasilense* to wheat when malate was added to the growth medium. Since, in the latter case nitrogen fixation is fuelled by exogenously added carbon rather than by the plant root exudate, the capacity for N-transfer is likely to be exaggerated.*

In conclusion, although we agree with the reviewer that $^{15}\text{N}_2$ transfer experiments provide a direct demonstration that dinitrogen gas is incorporated into a plant specific molecule, in our case the nitrogen contribution provided by BNF is likely to be significantly under-estimated as a consequence of plant enclosure in an artificial environment. We now allude to this issue in the last paragraph of the Results section.

(2) As already mentioned in the initial review, the authors must fix all instances of imprecise language that is confusing nitrogen contribution / incorporation / provision with experiments that measure nitrogen content or plant growth promotion (examples pointed out in detailed comments). An increase in nitrogen content or biomass does not mean it was provided via BNF – in fact isotope incorporation studies that actually measure nitrogen contribution show that a much lower amount of N was incorporated in this study. Imprecise language such as this is unfortunately common and has not been healthy for the BNF field: together we should aim for a higher standard.

We agree that measurement of nitrogen content or biomass does not imply that nitrogen is provided by BNF, but as noted above the $^{15}\text{N}_2$ gas incorporation experiments do not necessarily reflect the precise amount of N incorporated by the plant because of the artificial physiological environment imposed by incubating plants in a sealed environment. Because of these limitations we believe that the ^{15}N dilution experiments may be more representative of the actual N contribution.

In response to the reviewer's comment, we have softened conclusions regarding nitrogen provision/contribution as requested in the detailed comments.

Detailed comments:

Introduction is well written!

Some parts of the MS need references: e.g. introduction paragraphs between lines 91 and 114, and also results paragraph concerning NMR study. Please check other sections that were added between original MS submission and revisions that have not been referenced. *Appropriate references have now been added.*

Figure 1a caption does not match figure (top/bottom should be left/right?). *We have corrected the legend accordingly.*

The authors might consider putting SI figure 1 (GS structure showing point mutation) into the main manuscript. It is more informative than main figures 1b-1d currently. Ng *We agree with this suggestion and have moved the GS structure and location of the P95L substitution into the main Fig. 2 to replace previous panels d and e, which are now included as Supplementary Figure 1.*

Concerning the point that less ammonia is consumed by Ec424 vs Ec in figures 2c and 2e and accompanying text in results: it should be pointed out in the text that this could also be due to the differences in growth rates of the two strains. The faster growing strain (Ec) is expected to consume ammonia faster. Given this more obvious explanation (compared to BNF), the authors might want to consider whether the ammonia consumption data are worthy for the main text.

We have now pointed out that we cannot entirely rule out the possibility that these results reflect growth rate differences between the strains. As suggested, this data has now been moved out of the main text into Supplementary Figure 3

Regarding stability, what happens **after 72h** to NH₃ accumulation in SI figure 3?

We presume that the reviewer is concerned that after an extended period of incubation (later than 72 hours), suppressor mutations could arise in the population that result in reassimilation of the ammonia? However, we observe that NH₃ accumulation is stable over a period of 144 hours at 23C, suggesting that reassimilation does not occur under these conditions. As this data relates to stability, we have included this result as an additional panel in Supplementary Fig 10 (panel B), in which NH₃ accumulation is shown to remain on a plateau for 144 hr for both Ko424 and Ec424(pKU7824).

Line 321 states maximum ammonia excretion occurs at 23C. Figure 3f however shows it occurs at 20C. Please reconcile.

We apologise for the contradiction and now state in the main text that maximum excretion occurs at 20C for both E. coli and strain Ko424

The NMR data (SI figure 5) is very impressive and the authors might consider showcasing it (as well as other isotope labelling data) in the main text. This type of rigorous experiment is not often done and elevates this study from others.

We thank the reviewer for this positive comment and as suggested, have moved the NMR data from the Supplementary Info into the new Fig. 4 (panel e).

Section concerning “Biochemical properties of the P95L variant of glutamine synthetase” is weak because only the WT enzyme was successfully characterized. This paragraph does not add value to the manuscript unless data from SI figure 7B are better interpreted or further investigated. E.g. Line 371: please explain why the K_m of the P95L mutant could not be determined and how despite not being able to determine it, the K_m is “apparently substantially reduced”. Additionally, a minor point for line 374: substantially different from what? (Clarify sentence). Unless elucidated in revisions, the section should also conclude that we do not yet understand how this mutation works biochemically.

We do not agree that this section is weak, as we believe it is necessary to confirm that the temperature sensitivity of the mutant strains in vivo is a direct consequence of the substitution in the variant GS enzyme. Although our studies with the purified enzyme suggest it is more temperature sensitive than wild-type GS, it was not possible to approach this mechanistically, since we were unable to fit the variant enzyme curves to the Michaelis-Menten equation, particularly at the lower temperatures where the enzyme activity is very low. Comparison of panels A and B in Supplementary Fig. 7 reveals that the variant enzyme does not saturate at very high levels of glutamate in comparison with the wild-type, suggesting that the K_m for glutamate might be relatively high. We have addressed this in the text as follows:

“it was not possible to accurately determine K_m values at the different temperatures, possibly because the enzyme activities were low”

To address the minor point on previous line 374 we have added:

“the apparent K_m for NH_4^+ of the wild type enzyme did not appear to differ substantially from 20°C to 37°C”

In response to the reviewer’s final comment, the concluding sentence of this section now reads:

“Overall, these data suggest that the GS-P95L enzyme is more temperature sensitive than wild-type GS in vitro, although the mechanistic basis for temperature sensitivity is not resolved”.

Figure 4 panel f should come before panels d and e, as panels d and e are “sampled” from panel f; this way of presenting it is more intuitive to the reader.

For the reasons stipulated below, we have decided to remove the co-culture experiments with E. coli from the manuscript.

Lines 391/392: nitrogen provision has not been shown in this experiment – soften conclusions.

The text related to previous Fig 4 has been removed.

Paragraph starting at line 393: there is a serious issue with the E. coli co-culture experiments. Constant incubation at 23C could result in a lower final OD than an oscillating temperature condition because growth rates of both strains are severely slowed at the lower temperature. This is a more likely conclusion than nitrogen provision and must be addressed. The control that is needed to strengthen the current conclusions is a 30C coculture. If this also results in a lower OD than the oscillating 23/30C condition, then the current conclusion is indeed supported. Additionally, the reader is asking themselves why this oscillation experiment was not done with algae co-culture as the section started this way – a clarifying sentence might help explain this.

The reviewer’s suggestion to perform control experiments with the E. coli co-culture at 30C is indeed a very important point, although from our perspective it is not obvious how the recipient organism

would be able to maintain growth at this temperature if nitrogen supplied from the donor is restricted. However, in contrast to expectations, we were surprised to find there was a greater growth increase at 30C than with the oscillating temperature shifts, thus invalidating our conclusion.

However, there is another factor involved here that we did not initially take into consideration, associated with the efficient nitrogen scavenging ability of *E. coli*. The *E. coli* ammonium transporter AmtB has a very low K_m for ammonia, permitting exponential growth at concentrations as low as 4 μM ammonium (Kim et al (2012) Mol Syst Biol 8). However, the Ko424 strain is competent to excrete low levels of ammonia at 30C (~200 micromolar -see Fig. 3d), which in theory is sufficient to support exponential growth of *E. coli*. We considered investigating this possibility through the use of an amtB deletion mutant of *E. coli*, but according to the above publication this only reduces the growth rate at ammonium concentrations lower than 20 μM .

Since we have been unable to confirm that the oscillating temperature shift is more beneficial for promoting the growth of *E. coli*, we have removed all the *E. coli* co-culture experiments from the manuscript.

As the above *E. coli* result potentially calls into question the validity of the conclusion that diurnal switches in temperature are beneficial to plant growth, we have repeated the coculture experiments with maize plants under all three conditions (i.e. oscillating temperature, constant 23 C and constant 30C) and also performed an independent reiteration. In contrast to the plants exposed to oscillating temperatures, we see no significant difference in the dry weight or N content of maize grown at constant 30C when inoculated with Ko424 in comparison with the other controls (please see new Fig. 6 and supplementary Figure 15). Hence our conclusions with respect to the diurnal temperature shifts are substantiated for the plant growth experiments.

We have also repeated the coculture experiments with the alga on agar plates under the fluctuating 12 hr 30C and 23C regime and observe that the chlorophyll content of *Chlorella* adjacent to Ko424 is higher with the oscillating temperatures in comparison with constant 23C or constant 30C (Figure 5).

Line 429: The authors should highlight that those 400 stable colonies are 100% of colonies screened. Given the previous literature reports of instability of NH_3 excretion phenotypes, this is impressive. It will also be critically viewed and a second piece of data might be needed to show stability. For example: as pointed out earlier, was ammonia reassimilated after 72 h (SI figure 3)? If not, that would be very important to show. If yes, it calls into question the 400-colony experimental design.

We have now emphasised that **all** 400 colonies were screened. As noted above, we do not observe apparent reassimilation of the excreted NH_3 during an extended incubation period of 144 hours (Supplementary Fig.10 B). For the experimental design, we chose a 72 hr period between each subculture, which we consider appropriate given that there is no evidence for instability after longer incubation periods.

Line 443: concerning the 12% increase in DW compared to all controls: SI figure 12C shows that that it is not 12% across the board but differs depending on control group. Please clarify in the text exactly what comparison was made.

We have revised the text here to state that the 12% refers to comparison with wild-type *K oxytoca*.

Lines 481-484: authors have not shown nitrogen provision in this experiment and therefore language should be changed to reflect this (the words provision and contribution are not

precise here).

We have changed the text to refer to plant N content rather than N provision.

Lines 536-542: concerning the discussion of the oscillator paradigm and its comparison to synthetic biology oscillators: (1) please cite examples of synthetic biology oscillators for context. (2) As a larger point, I think this section of the discussion should be reframed. The distinction between this work and other oscillators is that this work has not actually built an oscillator. The oscillator is external (temperature), whereas synthetic biology often aims to build internal oscillators – which are much more complex. Instead, this work makes use of an existing external oscillator (temperature) and exploits the activity of an enzyme mutant as well as its phenotype in response to it. Additionally, the authors might want to consider combining this part of the discussion with the current last paragraph of the discussion that talks about other potentially oscillating natural switches.

(1) We have cited references as requested.

(2) We agree that we have not built an oscillator, but rather the oscillation is a consequence of the external temperature influencing the activity of a temperature sensitive enzyme with downstream metabolic and regulatory consequences. We have attempted to emphasise in the revision that the oscillation is a natural consequence of external temperature variation. However, we do not concur with the use of the term “internal” when referring to synthetic oscillators, because these can also be designed to respond to external signals. Moreover, although in our case the oscillation is ambient temperature-dependent, metabolic oscillation still occurs inside the cell. Instead, we have used the term “de novo” to distinguish synthetic oscillators, which we believe implies they are engineered novel circuits “built from scratch”.

We have now coupled this point to the regulations concerning release of genetically modified microorganisms in agriculture as single gene mutations are obviously preferable to more extensive engineering, particularly outside the US, where regulations are more stringent.

We feel that this part of the discussion fits best in its current location, so we have not merged it into the last paragraph.

Lines 579 and 581 and 599: 34% and 18% and 11-15% respectively are measuring nitrogen content, not nitrogen contribution. Please re-word to be precise. There are no data to support that this much nitrogen is being contributed by Ko424 in this paper.

We have amended the text accordingly to describe the increase in terms of nitrogen content.

Reviewers' Comments:

Reviewer #4:

Remarks to the Author:

The manuscript is largely improved and significantly more precise and rigorous. Specifically, language around growth promotion, nitrogen fixation, nitrogen content, and nitrogen contribution to plants has been clarified. Conclusions are properly stated in scope and alternative interpretations of the data mentioned. Previously questionable interpretations and data have been removed or further evidence has been brought in to substantiate them.

I can now – with excitement – recommend this paper for publication following adjustments of a few items (below) that will not require re-review if deemed satisfactorily implemented by the editor:

1) Most importantly: at the end of the results section, only the limits of the isotope incorporation experiment are discussed, but not the limits of the isotope dilution experiment. For balance and to avoid confirmation bias, isotope dilution experiment limits must be discussed and can include e.g. confounding by the difference in size of reference plant to plant of interest (size affects dilution) and that they don't study a plant-specific molecule. These and other factors can lead to inflated data. I propose a final sentence be added to bring both data sets together. E.g. something along the lines of: "Given data from isotope dilution and incorporation experiments, we have validated that fixed nitrogen is indeed transferred to plants; how much exactly is contributed between 1.5%-26% is difficult to conclude given limitations of both experiments."

2) Also important: Figure 5 as it stands right now does not show algae growth is due to the ammonia excreting mutant because controls are missing. Authors likely have these data already (some are in supplementary figure 9) and should expand the main figure. Specifically, valuable would be two sets of controls across 23C, 30C, and 23-30C oscillations: (1) algae growth with the WT Ko strain to show the effect the GS mutant has on algae growth vs the WT strain itself and (2) algae growth with nitrogen in the media to show that growth at 30C can be rescued and is not due to temperature sensitivity of the algae.

3) Please briefly address why extracellular ammonia accumulation in supplementary figure 10B plateaus after 72 h (meaning it stops). Why does it not keep increasing?

4) Supplementary figure 15 – please show individual data points like in main manuscript figure 6.

Dear Editor, in response to the reviewer's comments we have modified Figure 5 as requested and addressed the other comments. The text of our response is coloured in blue.

Reviewer #4 (Remarks to the Author):

The manuscript is largely improved and significantly more precise and rigorous. Specifically, language around growth promotion, nitrogen fixation, nitrogen content, and nitrogen contribution to plants has been clarified. Conclusions are properly stated in scope and alternative interpretations of the data mentioned. Previously questionable interpretations and data have been removed or further evidence has been brought in to substantiate them.

I can now – with excitement – recommend this paper for publication following adjustments of a few items (below) that will not require re-review if deemed satisfactorily implemented by the editor:

1) Most importantly: at the end of the results section, only the limits of the isotope incorporation experiment are discussed, but not the limits of the isotope dilution experiment. For balance and to avoid confirmation bias, isotope dilution experiment limits must be discussed and can include e.g. confounding by the difference in size of reference plant to plant of interest (size affects dilution) and that they don't study a plant-specific molecule. These and other factors can lead to inflated data. I propose a final sentence be added to bring both data sets together. E.g. something along the lines of: "Given data from isotope dilution and incorporation experiments, we have validated that fixed nitrogen is indeed transferred to plants; how much exactly is contributed between 1.5%-26% is difficult to conclude given limitations of both experiments."

We have addressed this as follows: "However, given the limitations of both the ^{15}N dilution and the ^{15}N incorporation techniques, it is difficult to precisely quantify the amount of nitrogen directly transferred to the plants via BNF".

2) Also important: Figure 5 as it stands right now does not show algae growth is due to the ammonia excreting mutant because controls are missing. Authors likely have these data already (some are in supplementary figure 9) and should expand the main figure. Specifically, valuable would be two sets of controls across 23C, 30C, and 23-30C oscillations: (1) algae growth with the WT Ko strain to show the effect the GS mutant has on algae growth vs the WT strain itself and (2) algae growth with nitrogen in the media to show that growth at 30C can be rescued and is not due to temperature sensitivity of the algae.

We have provided a new version of Fig 5 in which all the controls requested above are provided.

3) Please briefly address why extracellular ammonia accumulation in supplementary figure 10B plateaus after 72 h (meaning it stops). Why does it not keep increasing?

It does not increase because nitrogen fixation stops before 72 h and growth ceases (please see Supplementary Fig 4).

4) Supplementary figure 15 – please show individual data points like in main manuscript figure 6.

These are now included in the figure.